# RUNX2 isoform II protects cancer cells from ferroptosis and apoptosis by promoting PRDX2 expression in oral squamous cell carcinoma

**Junjun Huang[1], Rong Jia[1,2]\*, Jihua Guo[1,3]\***

[1]State Key Laboratory of Oral & Maxillofacial Reconstruction and Regeneration, Key Laboratory of Oral Biomedicine Ministry of Education, Hubei Key Laboratory of Stomatology, School & Hospital of Stomatology, Wuhan University, Wuhan, China; [2]RNA Institute, Wuhan University, Wuhan, China; [3]Department of Endodontics, School and Hospital of Stomatology, Wuhan University, Wuhan, China

## eLife Assessment

This article investigates how isoform II of transcription factor RUNX2 promotes cell survival and proliferation in oral squamous cell carcinoma cell lines. The authors used gain and loss of function techniques to provide **convincing** evidence showing that RUNX2 isoform silencing led to cell death via several mechanisms including apoptosis and ferroptosis that was partially suppressed through RUNX2 regulation of PRDX2 expression. The study provides **valuable** insight into the underlying mechanism by which RUNX2 acts in oral squamous cell carcinoma.

**\*For correspondence:**
jiarong@whu.edu.cn (RJ);
jihuaguo@whu.edu.cn (JG)

**Competing interest:** The authors declare that no competing interests exist.

**Abstract** Ferroptosis is a distinct iron-dependent programmed cell death and plays important roles in tumor suppression. However, the regulatory mechanisms of ferroptosis need further exploration. RUNT-related transcription factor 2 (RUNX2), a transcription factor, is essential for osteogenesis. *RUNX2* has two types of transcripts produced by two alternative promoters. In the present study, we surprisingly find that RUNX2 isoform II is a novel ferroptosis and apoptosis suppressor. RUNX2 isoform II can bind to the promoter of peroxiredoxin-2 (*PRDX2*), a ferroptosis inhibitor, and activate its expression. Knockdown of RUNX2 isoform II suppresses cell proliferation in vitro and tumorigenesis in vivo in oral squamous cell carcinoma (OSCC). Interestingly, homeobox A10 (HOXA10), an upstream positive regulator of RUNX2 isoform II, is required for the inhibition of ferroptosis and apoptosis through the RUNX2 isoform II/PRDX2 pathway. Consistently, RUNX2 isoform II is overexpressed in OSCC, and associated with OSCC progression and poor prognosis. Collectively, OSCC cancer cells can upregulate RUNX2 isoform II to inhibit ferroptosis and apoptosis and facilitate tumorigenesis through the novel HOXA10/RUNX2 isoform II/PRDX2 pathway.

## Introduction

Ferroptosis, a form of non-apoptotic programmed cell death, is an iron-dependent death and is characterized by the accumulation of lipid peroxidation and the production of reactive oxygen species (ROS) (*Dixon et al., 2012*). Ferroptosis has sparked great interest as targeting ferroptosis might provide new therapeutic opportunities in treating cancers. Emerging evidence indicates that ferroptosis may function as a potent tumor suppressor in the progression of head and neck squamous cell carcinoma (HNSCC) (*Li et al., 2022b*; *Lu et al., 2022*), including oral squamous cell

**eLife digest** Most tumors in the jaw, mouth, and face are caused by a type of cancer known oral squamous cell carcinoma (or OSCC for short). However, current therapies for OSCC are limited and new approaches are desperately needed.

One promising therapeutic target is the protein RUNX2, which has been shown to promote the progression of cancers, including OSCC. Most cells produce two slightly different versions of the RUNX2 protein, known as isoform I and isoform II. However, how these two isoforms contribute to tumor formation is poorly understood.

Huang et al. found that isoform II – but not isoform I – was highly abundant in the tissues of patients diagnosed with OSCC, particularly in individuals whose tumors were more developed and harder to treat. Further experiments revealed that this increased production is regulated by another protein called HOXA10, which specifically activates the gene that codes for isoform II.

The team discovered that OSCC cells need RUNX2 isoform II in order to multiply and grow in to tumors. Isoform II enhances cell proliferation by triggering the production of another protein called PRDX2, which blocks two carefully regulated programs of cell death: apoptosis and ferroptosis. In the absence of these two programs, OSCC cells are able to grow and expand without any restrictions, resulting in an overgrowth of cancerous tissue.

This work has not only identified a new pathway for tumor formation, but also a novel regulatory mechanism for ferroptosis and apoptosis. These findings suggest that members of this pathway – HOXA10, RUNX2 isoform II, and PRDX2 – could be new therapeutic targets for OSCC, and potentially other types of cancer that are also associated with high levels of RUNX2 isoform II.

carcinoma (OSCC) (*Sun et al., 2022*; *Yang et al., 2021*). Furthermore, triggering ferroptosis can overcome OSCC-acquired drug resistance such as cisplatin-induced resistance (*Han et al., 2022*). Therefore, the induction of ferroptosis is an attractive strategy for OSCC therapy. Multiple extrinsic or intrinsic pathways regulate the ferroptotic process (*Tang and Kroemer, 2020*). The extrinsic pathways are initiated by the inhibition of cell membrane transporters such as the cystine/glutamate transporter system xc⁻ or by the activation of the iron transporters transferrin (TF) and lactotransferrin (LTF). The intrinsic pathway is activated by the blockade of intracellular antioxidant enzymes like glutathione peroxidase 4 (GPX4). Although the principal regulatory mechanisms of ferroptosis have been explored in the past few years, other potential molecular mechanisms remain to be uncovered.

The RUNT-related transcription factor 2 (RUNX2), a member of the RUNT-related transcription factor family, is critical for osteogenesis (*Bruderer et al., 2014*) and has been extensively studied in the development of bone and tooth (*Wen et al., 2020*; *Ziros et al., 2008*). Previous studies revealed that RUNX2 promotes cancer metastasis and invasion in a variety of cancers, including breast cancer (*Chang et al., 2014*), thyroid cancer (*Sancisi et al., 2012*), colorectal cancer (*Ji et al., 2019*), prostate cancer (*Akech et al., 2010*), lung cancer (*Herreño et al., 2019*), and HNSCC (*Chang et al., 2016*). However, the potential roles of RUNX2 in other aspects of tumorigenesis remain largely unclear. *RUNX2* gene is composed of two promoters that in turn generate two types of isoforms, isoform II derived from promoter 1 (P1) and isoform I from promoter 2 (P2). These isoforms contain distinct N-terminal sequences. Whether these isoforms play distinct roles in tumorigenesis remains unknown.

OSCC is one of the most common malignant cancers in the world, especially in areas with low Human Development Index (*Sung et al., 2021*). The treatment of OSCC has progressed over time from surgery alone to comprehensive series of therapies, including radiation, chemotherapy, and immunotherapy (*Chi et al., 2015*; *Kang et al., 2015*). Despite a lot of advances in treatment modalities, the 5-year overall survival rate is around 50–60% (*Jäwert et al., 2021*). Therefore, it is still challenging to improve the therapy of OSCC.

In this study, to have a well-defined understanding of the relationship between RUNX2 isoform II and ferroptosis, we examined the effects of isoform II-knockdown or -overexpression on total ROS levels and lipid peroxidation, apoptosis, and the effects of isoform II-knockdown on mitochondrial morphology in OSCC cells. And mechanically, peroxiredoxin-2 (*PRDX2*), a ferroptosis suppressor, was identified to be a target gene of RUNX2 isoform II. Meanwhile, we analyzed the effect of RUNX2

isoform II overexpression or knockdown on OSCC cell ferroptosis or apoptosis, cell proliferation, and tumor growth.

## Results

### RUNX2 isoform II is overexpressed and associated with poor prognosis in OSCC

Given that the expression levels as well as the functions of RUNX2 isoforms (isoform I and II) produced by two alternative promoters (*Figure 1A*) in tumors are unclear, we explored their expression and roles in OSCC in this study. The expression levels of total RUNX2 were slightly higher in TCGA OSCC tissues than those in normal controls, but there was no statistically significant difference (*Figure 1B*). However, isoform I and II were significantly differently expressed in OSCC. The expression levels of isoform II (indicated by PSI, percent-splice-in, the usage of exon 1.1 in total *RUNX2* transcripts) in OSCC patients were 1.46-fold significantly higher than those in normal controls (*Figure 1C*). In contrast, the expression levels of isoform I were lower than those in normal controls (*Figure 1D*). Moreover, patients with clinical stage I, II, and III showed lower levels of isoform II compared with those with stage IV (*Figure 1E*). Patients with higher isoform II showed significantly shorter overall survival (*Figure 1F*), whereas patients with higher isoform I showed longer overall survival (*Figure 1— figure supplement 1*). These evidences suggested that isoform II was highly expressed in OSCC tissues and positively correlated with the progression of OSCC.

Consistently, the expression levels of isoform II were also significantly upregulated in breast invasive carcinoma (BRCA) (*Figure 1—figure supplement 2A*), colon adenocarcinoma (COAD) (*Figure 1— figure supplement 2B*), prostate adenocarcinoma (PRAD) (*Figure 1—figure supplement 2C*), and stomach adenocarcinoma (STAD) (*Figure 1—figure supplement 2D*) patients of TCGA database, suggesting that isoform II might play extensive roles in multiple cancers.

To verify the results from TCGA, we analyzed the expression levels of *RUNX2* isoforms in 11 OSCC tissues and adjacent normal controls by RT-PCR (*Figure 1G*). As expected, the ratios of isoform II vs I and the expression levels of isoform II were also significantly higher in these OSCC tissues than those in adjacent normal tissues (*Figure 1H and I*). These results suggested that RUNX2 isoform II may play important roles in OSCC.

### RUNX2 isoform II is required for OSCC cell proliferation in vitro and tumorigenesis in vivo

Then, we explored the roles of RUNX2 isoform II in OSCC cells. Overexpression of FLAG tagged isoform II significantly promoted the proliferation of OSCC cells, CAL 27 and SCC-9 cell lines (*Figure 2A and B*, *Figure 2—figure supplement 1A and B*). Interestingly, isoform II overexpression showed enhanced cell proliferation compared with isoform I overexpression in OSCC cells (*Figure 2A*). Consistently, knockdown of isoform II significantly inhibited cell proliferation in both cell lines (*Figure 2C and D*), as well as colony formation (*Figure 2E*). We found that isoform II overexpression has no effect on OSCC cell migration or invasion, while isoform I overexpression could inhibit the cell migration or invasion (*Figure 2—figure supplement 2A–C*). Cells with isoform II-knockdown showed significantly higher apoptosis than those control cells (*Figure 2F*). However, the overexpression of isoform II or isoform I had no obvious effect on the cellular apoptosis of OSCC (*Figure 2—figure supplement 3*). Importantly, CAL 27 cells stably transfected with shRNAs against isoform II showed significantly reduced tumor growth and weight than those transfected with non-specific control shRNA in nude mice (*Figure 2G–J*, *Figure 2—figure supplement 4A–D*). These results suggested that RUNX2 isoform II is required for the proliferation and tumorigenicity of OSCC cells.

### RUNX2 isoform II suppresses ferroptosis

Next, we explored how RUNX2 isoform II enhanced the proliferation of OSCC cells. Ferroptosis is an important form of programmed cell death and plays an important role in the suppression of tumors (*Ouyang et al., 2022*; *Wei et al., 2021*). We found that OSCC tissues had positive 4-hydroxynonenalince (4-HNE, a metabolite of lipid peroxidation) staining in variant levels, suggesting that ferroptosis might be present in OSCC tissues (*Figure 3—figure supplement 1*). Ferroptosis is characterized by the accumulation of lipid peroxidation and ROS. We found that isoform II-knockdown significantly enhanced

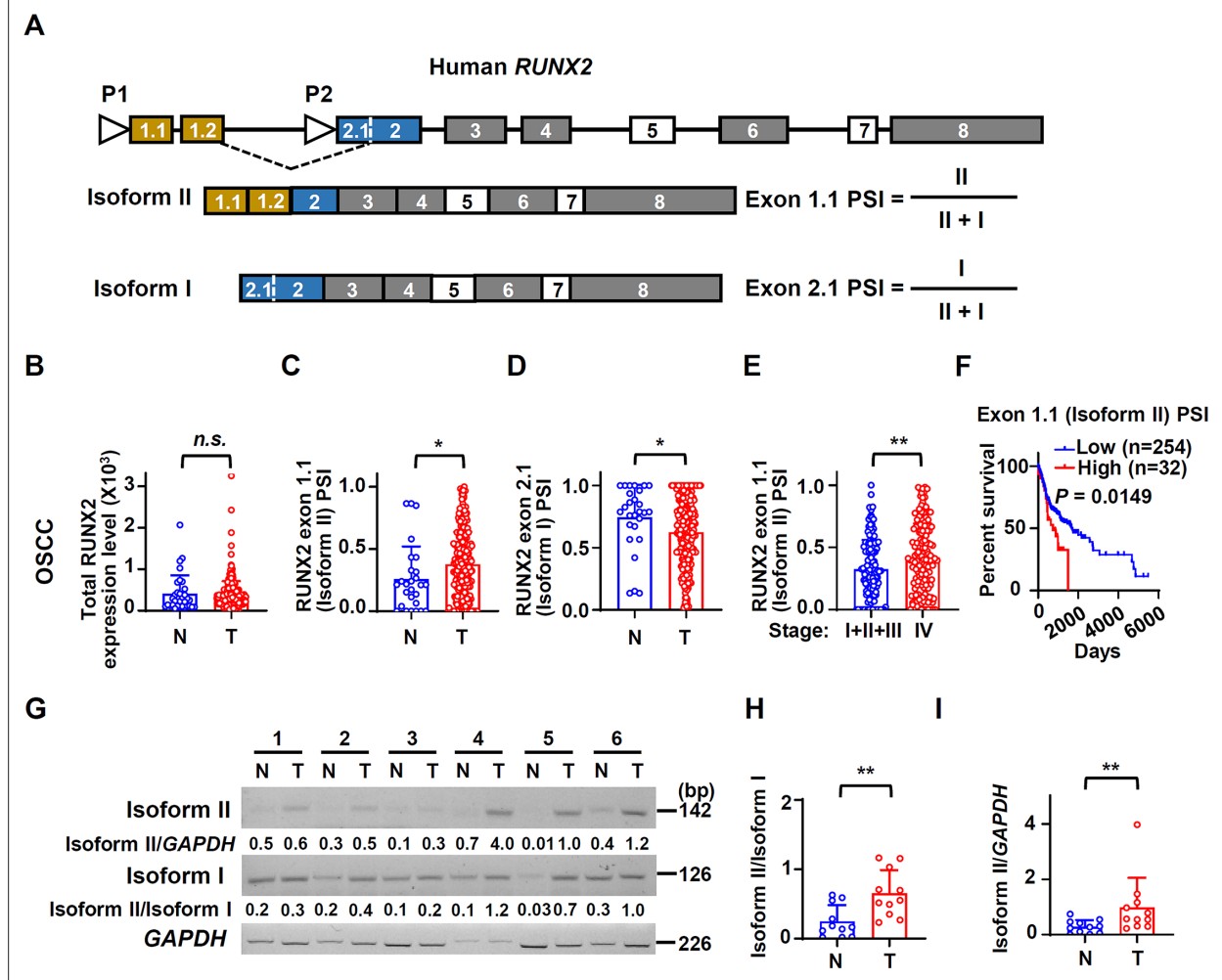

**Figure 1.** Human RUNX2 isoform II is overexpressed and associated with poor prognosis in oral squamous cell carcinoma (OSCC). (**A**) Schematic diagram of the isoforms and alternative promoters of the human *RUNX2* gene. Boxes and lines represent exons or introns in the pre-mRNA, respectively. P1 and P2 represent promoters. Isoform II is transcribed from P1, while isoform I is transcribed from P2. Exon 5 and 7 are alternative exons. (**B–D**) The expression levels of total RUNX2 and isoforms in TCGA OSCC patients. (**B**) The normalized expression levels of total RUNX2, obtained from an online program, TSVdb, in normal (32 cases) or OSCC tissues (309 cases). (**C, D**) The percent-splice-in (PSI) values of exon 1.1 (isoform II) (**C**) and exon 2.1 (isoform I) (**D**) (total 288 cases with PSI values of exon 1.1 and exon 2.1) in normal (27 cases) and OSCC tissues (288 cases) were obtained from an online program, TCGA SpliceSeq. The PSI values represent the relative expression levels of individual isoform. (**E**) Comparison of exon 1.1 (isoform II) PSI between patients in stage I, II and III (132 cases) and those in stage IV (156 cases) of OSCC TCGA patients. (**F**) OSCC TCGA patients (total 286 cases with survival data) with low expression (254 cases) or high expression (32 cases) of exon 1.1 (isoform II) in OSCC. Low exon 1.1 (isoform II) PSI was defined as less than mean + 1.427 SD. (**G**) The representative RT-PCR results of isoform II and isoform I in our OSCC or normal samples. *GAPDH* served as a loading control. (**H–I**) The scatter dot plot summarized the ratio of isoform II versus isoform I (isoform II/isoform I) (**H**) or the relative expression levels of isoform II (isoform II/*GAPDH*) (**I**) in our clinical OSCC (11 cases) and normal samples (11 cases). *p<0.05, **p<0.01.

The online version of this article includes the following source data and figure supplement(s) for figure 1:

**Source data 1.** Raw data files for *Figure 1B-E* and *Figure 1G-I*.

**Figure supplement 1.** Lower RUNX2 isoform I expression is associated with poor overall survival in oral squamous cell carcinoma (OSCC) patients.

**Figure supplement 2.** RUNX2 isoform II is also overexpressed in some other carcinomas.

**Figure supplement 2—source data 1.** Raw data files for *Figure 1—figure supplement 2A-D*.

total ROS production (*Figure 3A*) and lipid peroxidation accumulation (*Figure 3B*) in CAL 27 and SCC-9 cells. Consistently, isoform II overexpression suppressed ROS production (*Figure 3C*) and lipid peroxidation (*Figure 3D*) in these cells. In addition, we found that isoform II-knockdown induced shrunken mitochondria with vanished cristae with transmission electron microscopy (*Figure 3E*). This phenomenon, along with the above results of ROS production and lipid peroxidation accumulation

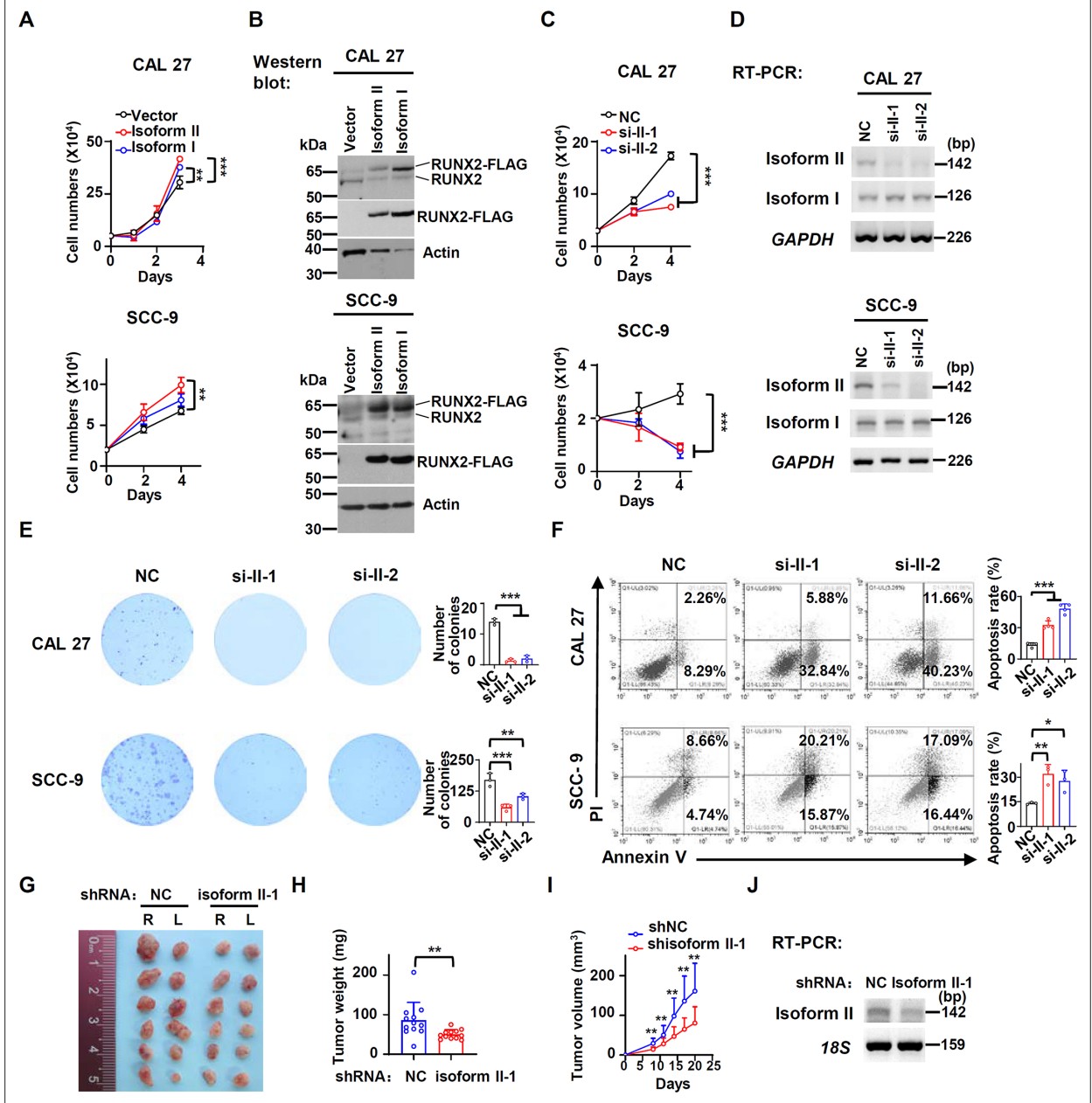

**Figure 2.** RUNX2 isoform II is required for the proliferation in vitro and tumorigenesis in vivo. (**A**) CAL 27 or SCC-9 cells were stably transfected by isoform II-expression, isoform I-expression or vector control lentivirus. CAL 27 cells were seeded into 24-well plates at day 0 and counted on day 1, 2, and 3. SCC-9 cells were seeded into 24-well plates at day 0 and counted on day 2 and 4. Data are means ± SD, n = 3. (**B**) Overexpression of RUNX2 isoform II or isoform I was confirmed by western blot. Actin served as a loading control. (**C**) Proliferation curves of CAL 27 and SCC-9 cells treated with anti-isoform II siRNAs (si-II-1 or si-II-2) or negative control siRNA (NC). Data are means ± SD, n = 3. (**D**) Knockdown efficiency of isoform II was analyzed by RT-PCR. *GAPDH* served as a loading control. (**E**) Effects of isoform II-knockdown on the clonogenic ability in CAL 27 and SCC-9. The histograms on the right summarized the numbers of colonies (at least 50 cells/colony). Data are means ± SD, n = 3. (**F**) CAL 27 and SCC-9 cells were treated with si-II-1, si-II-2, or NC siRNA. The cellular apoptosis was analyzed by flow cytometry. The histograms on the right summarized the cellular apoptosis. Data are means ± SD, n=4 for CAL 27, n = 3 for SCC-9. (**G–I**) CAL 27 cells with stable isoform II shRNA (shisoform II-1) or nonspecific shRNA (shNC) were injected into both sides of the dorsum of BALB/c nude mice. (**G, H**) Tumors were dissected out and weighed on day 21. (**I**) Tumor volumes were measured on different days. (**J**) Knockdown efficiency of isoform II was analyzed by RT-PCR. *18S* rRNA served as a loading control. *p<0.05, **p<0.01, ***p<0.001.

The online version of this article includes the following source data and figure supplement(s) for figure 2:

**Source data 1.** Raw data files for *Figure 2B, D-F, H and J*.

**Figure supplement 1.** RUNX2 isoform II overexpression promoted cell proliferation.

*Figure 2 continued on next page*

*Figure 2 continued*

**Figure supplement 1—source data 1.** Raw data files for *Figure 2—figure supplement 1B*.

**Figure supplement 2.** RUNX2 isoform II has no effect on migration or invasion of oral squamous cell carcinoma (OSCC) cells.

**Figure supplement 2—source data 1.** Raw data files for *Figure 1—figure supplement 2A-C*.

**Figure supplement 3.** Overexpression of RUNX2 isoform II or isoform I did not affect cellular apoptosis.

**Figure supplement 3—source data 1.** Raw data file for *Figure 2—figure supplement 3*.

**Figure supplement 4.** RUNX2 isoform II-knockdown inhibits the tumor growth.

**Figure supplement 4—source data 1.** Raw data files for *Figure 2—figure supplement 4B and D*.

assays, suggests that RUNX2 isoform II may suppress ferroptosis. We also found that isoform II-knockdown cells showed more elongated mitochondria than in control cells (*Figure 3E*, *Figure 3—figure supplement 2A*), which may be related to the suppression of *FIS1* (Fission, Mitochondrial 1) expression by RUNX2 isoform II-knockdown (*Figure 3—figure supplement 2B*). And the elongated mitochondria may be caused by the deficiency of mitochondrial fission (*Wang et al., 2024*). The oxygen consumption rates (OCRs) were lower in isoform II-knockdown OSCC cells (*Figure 3—figure supplement 3A and B*). To further figure out whether isoform II could promote cell proliferation by inhibiting cell death, especially ferroptosis, and apoptosis, we performed the rescue experiments with some inhibitors of cell death including ferrostatin-1 (Fer-1, a ferroptosis inhibitor), Z-VAD (an apoptosis inhibitor), and necrostatin-1 (Nec-1, a necroptosis inhibitor) upon isoform II-knockdown. As expected, Fer-1 treatment partially rescued the reduction of cell proliferation caused by isoform II-knockdown (*Figure 3F*, *Figure 3—figure supplement 4A*). We also found that Z-VAD and Nec-1 could partially rescue the reduction of cell proliferation caused by isoform II-knockdown (*Figure 3G*, *Figure 3—figure supplement 4B*) and Z-VAD also could partially decrease apoptosis induced by isoform II-knockdown (*Figure 3—figure supplement 5*), which suggested that both apoptosis and necroptosis also contribute to the retardation of cell proliferation cause by isoform II-knockdown. In addition, we found that the combination of Fer-1 and Z-VAD was more effective in rescuing cell proliferation than Fer-1 or Z-VAD alone (*Figure 3—figure supplement 6*). Fer-1 treatment reduced the increased levels of ROS production (*Figure 3H*) and lipid peroxidation (*Figure 3I*) caused by isoform II-knockdown. RSL3, a ferroptosis activator, could cause cell death in CAL 27 and SCC-9 cells (*Figure 3—figure supplement 7A*), and increase the production of cellular ROS (*Figure 3—figure supplement 7B*) and lipid peroxidation (*Figure 3—figure supplement 7C*). Isoform II overexpression could elevate the $IC_{50}$ values of RSL3 (*Figure 3—figure supplement 8A*); in contrast, isoform II-knockdown decreased the $IC_{50}$ values of RSL3 (*Figure 3—figure supplement 8B*). As expected, overexpression of isoform II could partially reduce the increased levels of ROS production (*Figure 3J*) and lipid peroxidation (*Figure 3K*) caused by RSL3. Consistently, tumors formed by CAL 27 cells with isoform II-knockdown showed a significantly increased staining of 4-HNE compared with the control (*Figure 3L*). In summary, these results suggested that RUNX2 isoform II can suppress ferroptosis in OSCC cells.

## RUNX2 isoform II promotes the expression of PRDX2

To understand the regulatory mechanisms of how RUNX2 isoform II suppresses ferroptosis, we screened some ferroptosis-suppressive genes including several antioxidant enzymes in CAL 27 treated with isoform II siRNAs. Firstly, we found that RUNX2 isoform II-knockdown or overexpression could downregulate or upregulate the expression of *GPX4* mRNA and protein, respectively (*Figure 4—figure supplement 1A–D*). In addition to the GPX4, we found that *PRDX2* is the most significantly downregulated gene upon isoform II-knockdown in CAL 27 (*Figure 4A*). Furthermore, both mRNA and protein expression levels of PRDX2 were reduced in CAL 27 and SCC-9 cells with isoform II-knockdown (*Figure 4B and C*). Consistently, tumors formed by CAL 27 cells with isoform II-knockdown also showed a significantly reduced expression of PRDX2 compared with the control (*Figure 4D*). Isoform II overexpression increased *PRDX2* mRNA and protein expression (*Figure 4E and F*). Whereas isoform I overexpression has no significant effect on PRDX2 expression (*Figure 4E and F*). In line with the inhibitory effect of isoform II-knockdown on tumor growth, CAL 27 cells stably transfected with anti-PRDX2 shRNAs showed notably reduced tumor growth and weight than those transfected with non-specific control shRNA in nude mice (*Figure 4—figure supplement 2A–D*). These results indicated that PRDX2 was a target of RUNX2 isoform II. One of the important characteristics of ferroptosis is the

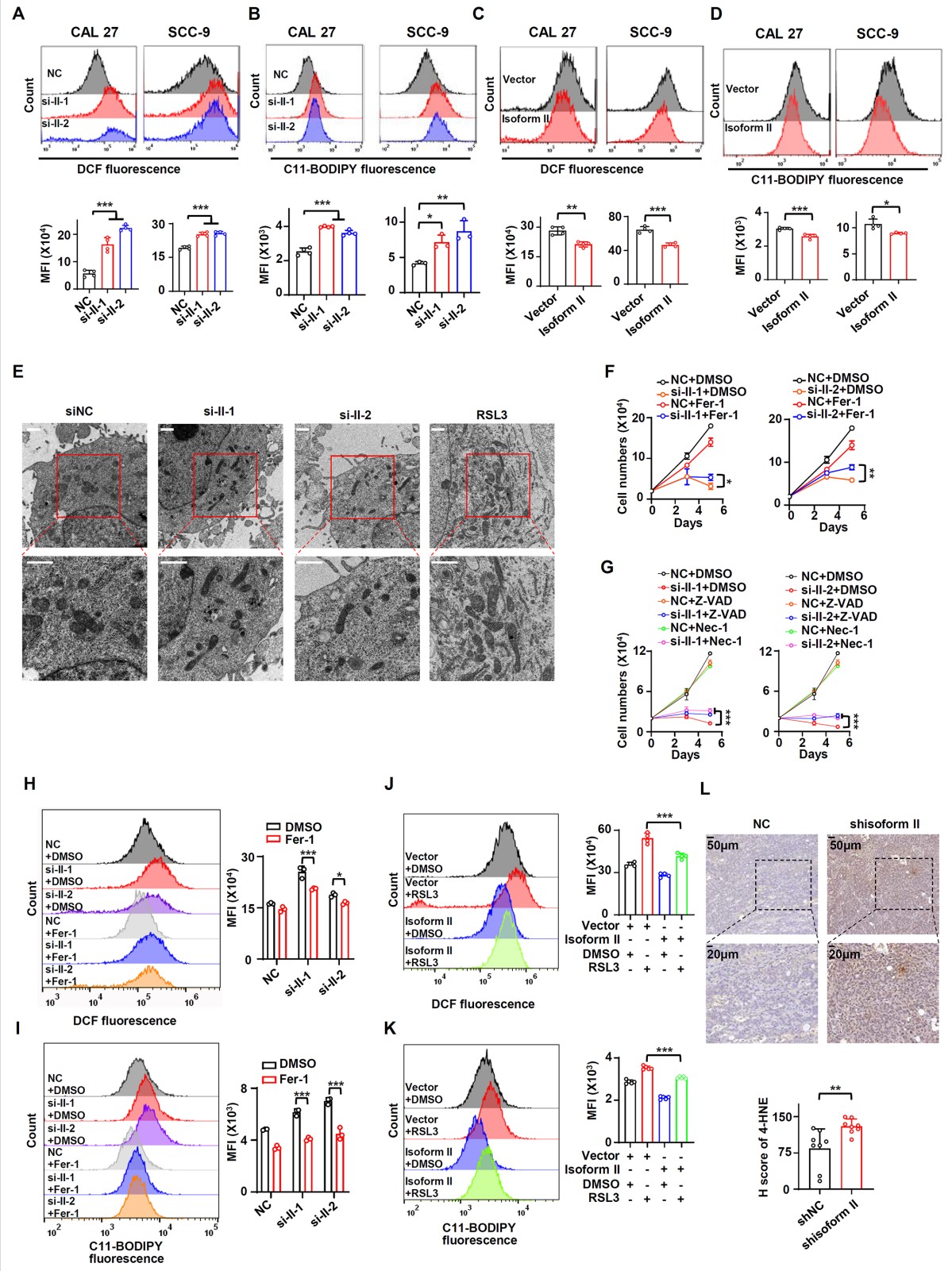

**Figure 3.** RUNX2 isoform II suppresses ferroptosis. (**A, B**) CAL 27 and SCC-9 cells were treated with anti-isoform II siRNAs (si-II-1 and si-II-2) or negative control siRNA (NC). (**A**) The levels of total reactive oxygen species (ROS) were detected with DCFH-DA using flow cytometry. The histograms below summarized the levels of mean fluorescent intensity (MFI). Data are means ± SD, n = 4. (**B**) The lipid peroxidation of cells was analyzed with C11 BODIPY 581/591 reagent using flow cytometry. The histograms below summarized the levels of MFI. Data are means ± SD, n = 4 for CAL 27, n = 3 for SCC-9. (**C,**

*Figure 3 continued on next page*

*Figure 3 continued*

D) CAL 27 or SCC-9 cells were stably transfected by isoform II-expression or vector control lentivirus. The levels of ROS (**C**) or lipid peroxidation (**D**) were detected by flow cytometry. The histograms below summarized the levels of MFI. Data are means ± SD, n = 4. (**E**) Transmission electron microscopy images of CAL 27 cells transfected with si-II-1, si-II-2, or NC. RSL3 (a ferroptosis activator) served as a positive control. Scale bar: 1 μm (**F**) CAL 27 cells transfected with anti-isoform II siRNAs were also treated with ferrostatin-1 (Fer-1, 10 μM), a ferroptosis inhibitor. Negative control siRNA and DMSO were used as controls. Cells were divided into six groups: NC + DMSO, si-II-1 + DMSO, si-II-2 + DMSO, NC + Fer-1, si-II-1 + Fer-1, and si-II-2 + Fer-1. To display clear diagrams, the proliferation curves of si-II-1-treated cells or si-II-2-treated cells were shown separately. Data are means ± SD, n = 3. (**G**) CAL 27 cells transfected with anti-isoform II siRNAs were also treated with Z-VAD (20 μM, an apoptosis inhibitor) or necrostatin-1 (Nec-1, 20 μM, a necroptosis inhibitor). Negative control siRNA and DMSO were used as controls. Cells were divided into nine groups: NC + DMSO, si-II-1 + DMSO, si-II-2 + DMSO, NC + Z-VAD, si-II-1 + Z-VAD, si-II-2 + Z-VAD, NC + Nec-1, si-II-1 +Nec-1, and si-II-2 + Nec-1. To display clear diagrams, the proliferation curves of si-II-1-transfected cells or si-II-2-transfected cells were shown separately. Data are means ± SD, n = 3. (**H, I**) The total ROS levels (**H**) or lipid peroxidation (**I**) of cells simultaneously transfected with anti-isoform II siRNAs or NC and treated with Fer-1 or DMSO were detected with DCFH-DA (**H**) or BODIPY 581/591 reagent (**I**) by flow cytometry. The histogram on the right summarized the levels of MFI. Data are means ± SD, n = 3. (**J, K**) The total ROS levels (**J**) or lipid peroxidation (**K**) of isoform II-overexpressed cells treated with RSL3 (2 μM, a ferroptosis activator) or DMSO were detected with DCFH-DA (**J**) or BODIPY 581/591 reagent (**K**) by flow cytometry. The histogram on the right summarized the levels of MFI. Data are means ± SD, n = 4 for (**J**), n = 5 for (**K**). (**L**) Representative images of immunohistochemical staining of 4-HNE in tumors with or without isoform II-knockdown (shisoform II vs shNC) in *Figure 2G*. The histogram below summarized the H score of 4-HNE staining in tumors. *p<0.05, **p<0.01, ***p<0.001.

The online version of this article includes the following source data and figure supplement(s) for figure 3:

**Source data 1.** Raw data files for *Figure 3A-D* and *Figure 3H-L*.

**Figure supplement 1.** Ferroptosis may be present in oral squamous cell carcinoma (OSCC) tissues.

**Figure supplement 2.** RUNX2 isoform II knockdown cells show more elongated mitochondria.

**Figure supplement 2—source data 1.** Raw data files for *Figure 3—figure supplement 2A and B*.

**Figure supplement 3.** RUNX2 isoform II-knockdown inhibits oxygen consumption rate (OCR) in oral squamous cell carcinoma (OSCC) cells.

**Figure supplement 3—source data 1.** Raw data files for *Figure 3—figure supplement 3B*.

**Figure supplement 4.** Validation of RUNX2 isoform II-knockdown in *Figure 3*.

**Figure supplement 4—source data 1.** Raw data files for *Figure 3—figure supplement 4A and B*.

**Figure supplement 5.** The apoptosis inhibitor Z-VAD reduces the increased apoptosis rates caused by isoform II-knockdown.

**Figure supplement 5—source data 1.** Raw data file for *Figure 3—figure supplement 5*.

**Figure supplement 6.** The combination of Fer-1 and Z-VAD partially rescues the deduced cell proliferation caused by isoform II knockdown.

**Figure supplement 6—source data 1.** Raw data file for *Figure 3—figure supplement 6*.

**Figure supplement 7.** Oral squamous cell carcinoma (OSCC) cell lines are sensitive to RSL3 treatment.

**Figure supplement 7—source data 1.** Raw data files for *Figure 3—figure supplement 7B and C*.

**Figure supplement 8.** Half maximal inhibitory concentration ($IC_{50}$) of RSL3 was analyzed in isoform II-overexpression CAL 27 (**A**) or in isoform II-knockdown CAL 27 (**B**).

imbalance in iron homeostasis, and iron transporter transferrin receptor (TFRC) has an important role in maintaining iron homeostasis. We found that RUNX2 isoform II-knockdown in OSCC cells had no obvious effect on the expression of TFRC (*Figure 4—figure supplement 3A and B*). And the expression level and localization of TFRC did not change in the tumors formed by CAL 27 with or without isoform II-knockdown (*Figure 4—figure supplement 3C*). These results indicated that RUNX2 isoform II might not regulate the cellular transport of iron.

To explore the regulatory mechanisms of how RUNX2 isoform II promotes PRDX2 expression, we applied the JASPAR to predict possible binding sites of RUNX2 on the *PRDX2* promoter. We analyzed 0–2440 bp upstream regions of the *PRDX2* transcription start site and found six potential binding sites for RUNX2 on the *PRDX2* promoter (*Figure 4G*). Chromatin immunoprecipitation and quantitative PCR (ChIP-qPCR) assay showed that RUNX2 isoform II could specifically bind to the *PRDX2* promoter (*Figure 4H and I*) and the amplified region is the base represented by the red box (*Figure 4G*). These results suggested that isoform II could bind to the *PRDX2* promoter and transactivate PRDX2 expression.

To further verify whether PRDX2 mediated the effect of isoform II on ferroptosis, we stably overexpressed FLAG-tagged PRDX2 in CAL 27 cells. We found that overexpression of PRDX2 could partially reduce the elevated cellular ROS levels (*Figure 4J*) and lipid peroxidation levels (*Figure 4K*) induced by isoform II-knockdown in CAL 27 (*Figure 4L and M*). In addition, we found that PRDX2 overexpression

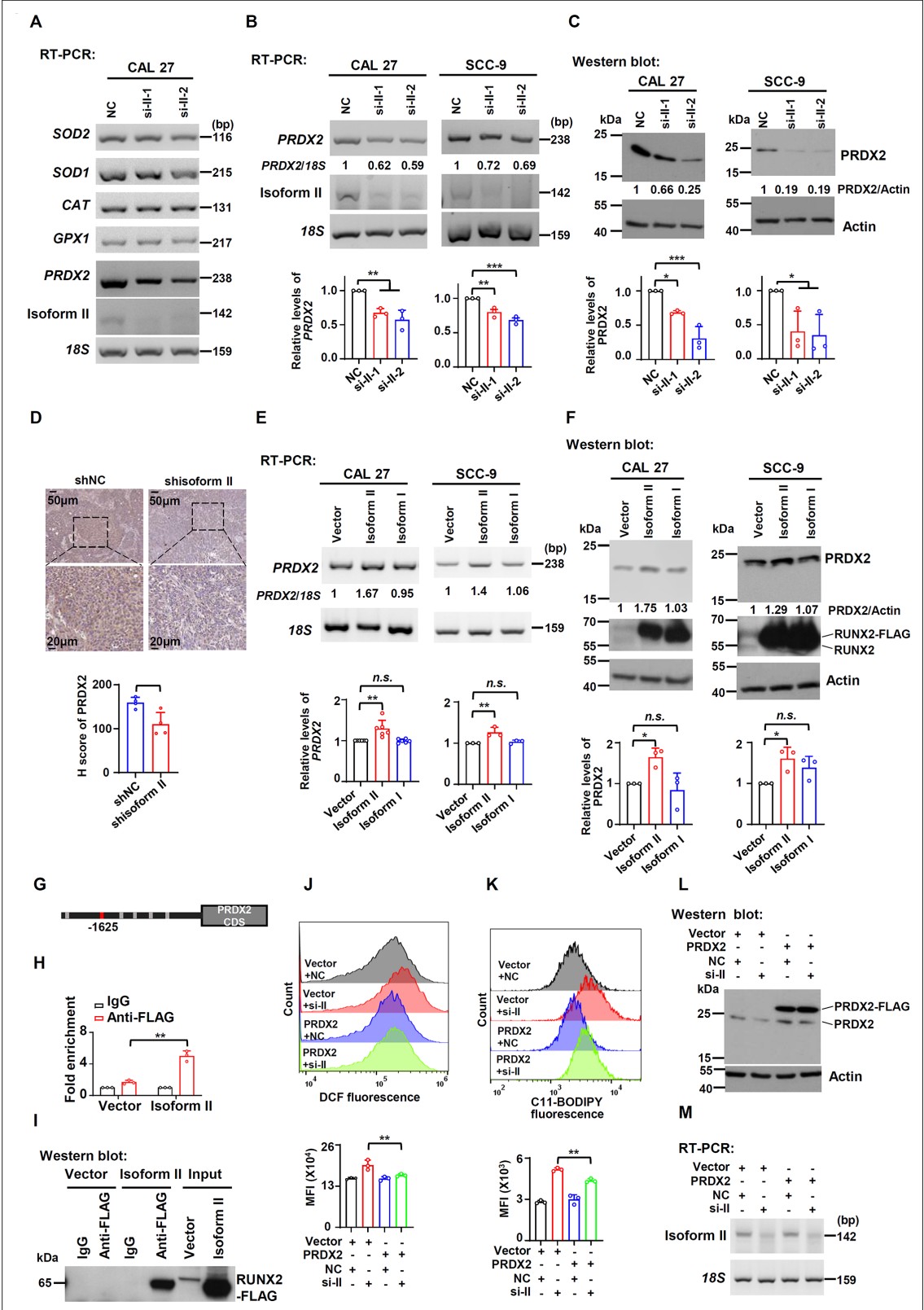

**Figure 4.** RUNX2 isoform II promotes the expression of PRDX2. (**A**) Screening analysis of the expression of enzymatic antioxidant genes upon isoform II-knockdown (si-II-1 and si-II-2) in CAL 27 cells *via* RT-PCR. (**B, C**) Effects of RUNX2 isoform II-knockdown on PRDX2 expression levels were analyzed by RT-PCR (**B**) or western blot (**C**) in CAL 27 or SCC-9. *18* S rRNA (**B**) or actin (**C**) served as loading controls. Data are means ± SD, n = 3. (**D**) Representative images of immunohistochemical staining of PRDX2 in tumors with or without isoform II-knockdown (shisoform II vs shNC) in *Figure 2G*. The histogram

*Figure 4 continued on next page*

*Figure 4 continued*

below summarized the expression levels of PRDX2 in tumors. (**E, F**) Effects of RUNX2 isoform II or isoform I overexpression on PRDX2 expression levels were analyzed by RT-PCR (**E**) or western blot (**F**) in CAL 27 or SCC-9. *18S* rRNA (**E**) or actin (**F**) served as loading controls. Data are means ± SD, n = 6 or 3 for CAL 27, n = 3 for SCC-9. (**G**) RUNX2 binding motifs on *PRDX2* promoter were obtained from JASPAR. (**H**) Chromatin immunoprecipitation and quantitative PCR (ChIP-qPCR) assay was performed in CAL 27 with or without FLAG-tagged RUNX2 isoform II overexpression (isoform II vs vector) by using anti-FLAG or control IgG antibody. Data are means ± SD, n = 3. (**I**) The immunoprecipitated protein levels of FLAG-tagged RUNX2 isoform II in the ChIP assays were determined by western blot. (**J–M**) CAL 27 cells were co-transfected with PRDX2-expression, empty control lentivirus, and anti-isoform II siRNA (si-II), negative control siRNA (NC). Transfected cells were divided into four groups: Vector + NC, Vector + si-II, PRDX2 + NC and PRDX2 + si-II. (**J, K**) The total reactive oxygen species (ROS) levels (**J**) or lipid peroxidation (**K**) of transfected cells were detected with DCFH-DA (**J**) or BODIPY 581/591 reagent (**K**) by flow cytometry. The histograms below summarized the levels of mean fluorescent intensity (MFI). Data are means ± SD, n = 3. (**L**) Overexpression of PRDX2 was confirmed by western blot. Actin served as a loading control. (**M**) Knockdown efficiency of isoform II was analyzed by RT-PCR. *18S* rRNA served as a loading control. *p<0.05, **p<0.01, ***p<0.001.

The online version of this article includes the following source data and figure supplement(s) for figure 4:

**Source data 1.** Raw data files for *Figure 4A-F* and *Figure 4H-M*.

**Figure supplement 1.** RUNX2 isoform II promotes the expression of GPX4 in oral squamous cell carcinoma (OSCC) cell lines.

**Figure supplement 1—source data 1.** Raw data files for *Figure 4—figure supplement 1A-D*.

**Figure supplement 2.** PRDX2-knockdown inhibits the tumor growth.

**Figure supplement 2—source data 1.** Raw data files for *Figure 4—figure supplement 2B and D*.

**Figure supplement 3.** RUNX2 isoform II has no effect on TFRC expression levels and localization in oral squamous cell carcinoma (OSCC) cells.

**Figure supplement 3—source data 1.** Raw data files for *Figure 4—figure supplement 3A-C*.

**Figure supplement 4.** PRDX2 overexpression rescues the apoptosis induced by isoform II knockdown.

**Figure supplement 4—source data 1.** Raw data file for *Figure 4—figure supplement 4*.

could partially reduce the increased apoptosis caused by isoform II-knockdown (*Figure 4—figure supplement 4*). These data indicated that isoform II-knockdown promoted ferroptosis or apoptosis through inhibiting PRDX2 expression.

## HOXA10 is required for RUNX2 isoform II expression and cell proliferation in OSCC

Next, we tried to understand the molecular mechanism of RUNX2 isoform II overexpression in OSCC cells. HOXA10 is an oncogenic transcription factor (*Guo et al., 2018*; *Song et al., 2019*). Mouse HOXA10 has been reported to bind to *Runx2* P1 promoter and then activate Runx2 isoform II expression in mouse cells (*Hassan et al., 2007*). Therefore, we speculated that RUNX2 isoform II overexpression in OSCC may be also caused by HOXA10. Indeed, HOXA10 knockdown significantly reduced isoform II expression in both CAL 27 and SCC-9 cells (*Figure 5A*), whereas isoform I expression was not significantly affected (*Figure 5—figure supplement 1A*). In line with the function of isoform II in OSCC cells, HOXA10 knockdown also significantly suppressed cell proliferation and colony formation (*Figure 5B and C*), and increased cellular apoptosis (*Figure 5D*). The expression levels of HOXA10 in TCGA OSCC patients were also significantly higher than those in normal controls (*Figure 5—figure supplement 1B*). Similarly, *HOXA10* expression level of our clinical OSCC tissues is significantly higher than that of adjacent normal tissues (*Figure 5—figure supplement 2A and B*). Moreover, TCGA OSCC patients with higher expression levels of HOXA10 showed shorter overall survival (*Figure 5—figure supplement 2C*). Consistently, the expression levels of isoform II were positively correlated with HOXA10 expression levels in TCGA OSCC patients (*Figure 5E*). These results suggested that HOXA10 can promote RUNX2 isoform II expression.

## HOXA10 inhibits ferroptosis and apoptosis through RUNX2 isoform II

Since RUNX2 isoform II was a ferroptosis and apoptosis suppressor, we speculated that HOXA10 could act as a ferroptosis and apoptosis inhibitor through upregulating the expression of isoform II. We found that the expression levels of *PRDX2* mRNA and protein significantly decreased in parallel with the reduction in isoform II expression caused by HOXA10 knockdown (*Figure 6A and B*). In addition, HOXA10-knockdown could suppress the expression of *GPX4* mRNA and protein (*Figure 6—figure supplement 1A and B*). Importantly, HOXA10 knockdown increased cellular ROS production (*Figure 6C*) and lipid peroxidation (*Figure 6D*) in CAL 27 and SCC-9 cells. Moreover,

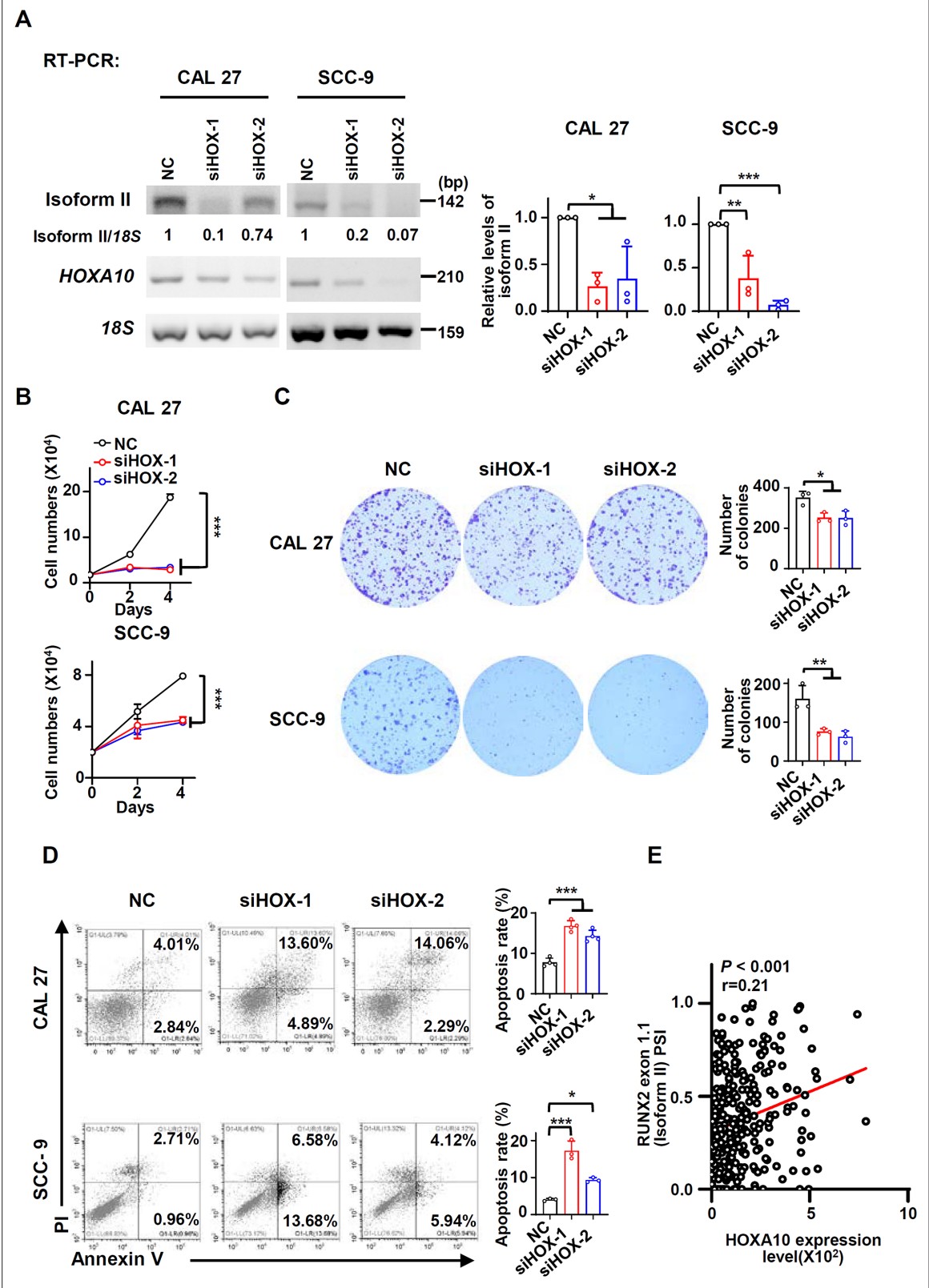

**Figure 5.** HOXA10 is required for RUNX2 isoform II expression and cell proliferation in oral squamous cell carcinoma (OSCC). (**A**) Effects of HOXA10 knockdown (siHOX-1 and siHOX-2) on isoform II expression levels were analyzed by RT-PCR in CAL 27 or SCC-9. *18S* rRNA served as a loading control. Data are means ± SD, n = 3. (**B–D**) CAL 27 or SCC-9 cells were treated with siHOX-1, siHOX-2, or NC siRNA. (**B**) Proliferation curves of CAL 27 or SCC-9 cells treated with HOXA10 siRNAs or NC siRNA. Data are means ± SD, n = 3. (**C**) Effects of HOXA10 knockdown on the clonogenic ability in CAL 27

*Figure 5 continued on next page*

*Figure 5 continued*

and SCC-9. The histograms on the right summarized the numbers of colonies (at least 50 cells/colony). Data are means ± SD, n = 3. (**D**) The cellular apoptosis was analyzed by flow cytometry. The histograms on the right summarized the cellular apoptosis. Data are means ± SD, n = 4 for CAL 27, n = 3 for SCC-9. (**E**) The expression of HOXA10 in TCGA OSCC patients (288 cases) is positively correlated with RUNX2 exon 1.1 (isoform II) PSI (Spearman's rank correlation coefficient, *r* = 0.21, p<0.001). *p<0.05, **p<0.01, ***p<0.001.

The online version of this article includes the following source data and figure supplement(s) for figure 5:

**Source data 1.** Raw data files for *Figure 5A, C and D*.

**Figure supplement 1.** HOXA10 does not affect the expression of isoform I.

**Figure supplement 1—source data 1.** Raw data files for *Figure 5—figure supplement 1A and B*.

**Figure supplement 2.** HOXA10 is overexpressed in oral squamous cell carcinoma (OSCC) and associated with poor overall survival.

**Figure supplement 2—source data 1.** Raw data files for *Figure 5—figure supplement 2A and B*.

Fer-1, a ferroptosis inhibitor, could partially rescue the retarded cell proliferation caused by HOXA10 knockdown (*Figure 6E*, *Figure 6—figure supplement 2A*). We also found that Z-VAD (an apoptosis inhibitor) (*Figure 6F*, *Figure 6—figure supplement 2B*), Nec-1 (a necroptosis inhibitor) (*Figure 6F*, *Figure 6—figure supplement 2B*) could partially rescue the reduction of cell proliferation, and Z-VAD could also partially rescue the elevated apoptosis induced by HOXA10 knockdown (*Figure 6—figure supplement 3*). The combination of Fer-1 and Z-VAD was more effective in rescuing cell proliferation than Fer-1 or Z-VAD alone (*Figure 6—figure supplement 4*). Fer-1 could partially decrease the cellular ROS levels (*Figure 6G*) and lipid peroxidation (*Figure 6H*) caused by HOXA10-knockdown. Stable overexpression of isoform II could partially rescue the retarded cell proliferation caused by HOXA10 knockdown (*Figure 7A*). The increased cellular apoptosis (*Figure 7B*), ROS production levels (*Figure 7C*) and lipid peroxidation levels (*Figure 7D*) caused by HOXA10 knockdown were also reduced in these OSCC cells (*Figure 7—figure supplement 1A and B*). Collectively, these results demonstrated that HOXA10 is required for OSCC cell proliferation by increasing RUNX2 isoform II expression, and decreasing ferroptosis and apoptosis.

In addition, isoform II overexpression could partially rescue the expression of *PRDX2* in both mRNA and protein levels in OSCC cells treated with anti-HOXA10 siRNA (*Figure 7E and F*). In addition, we also found that PRDX2 overexpression could partially decrease the cellular ROS levels (*Figure 7G*) and lipid peroxidation levels (*Figure 7H*) induced by HOXA10 knockdown (*Figure 7—figure supplement 1C and D*). PRDX2 overexpression also could rescue the increased cellular apoptosis caused by HOXA10 knockdown (*Figure 7—figure supplement 2*). These data showed that HOXA10 knockdown promoted the ferroptosis and apoptosis in OSCC cells, partially through PRDX2, a downstream target of isoform II.

## Discussion

Ferroptosis is a new form of programmed cell death and is caused by massive lipid peroxidation-mediated membrane damage (*Stockwell et al., 2017*). Emerging evidence has proved that ferroptosis contributes to the suppression of tumor progression. p53 could suppress the transcription of amino acid antiporter solute carrier family 7 member 11 (SLCA711) to sensitize cells to ferroptosis, which may contribute to the anti-tumor role of p53 (*Jiang et al., 2015*). Tumor inhibition by BRCA1-associated protein 1 (BAP1) could be achieved in part by inhibiting SLC7A11 and thereby promoting ferroptosis (*Zhang et al., 2018*). In HNSCC, Inhibition of xCT could suppress cell proliferation by inducing ferroptosis (*Li et al., 2022b*). In addition, caveolin-1 (CAV-1) also could promote HNSCC progression through inhibiting ferroptosis (*Lu et al., 2022*). Therefore, enhancing ferroptosis could be an attractive strategy for OSCC treatment. In this study, we discovered a novel ferroptosis suppressor, RUNX2 isoform II, in OSCC. Isoform II overexpression or knockdown inhibited or promoted OSCC cell ferroptosis by decreasing or increasing total ROS levels and lipid peroxidation, respectively. One of the characteristics of ferroptosis is elevated cellular ROS levels, thus ferroptosis can be modulated by antioxidants (*Tang et al., 2021*). For example, peroxiredoxin-6 (PRDX6) is a negative regulator of ferroptosis (*Lu et al., 2019*). In this study, we identified a new target gene for isoform II, the ferroptosis suppressor *PRDX2*. PRDX2 is a typical 2-Cys antioxidant enzyme belonging to the peroxiredoxin family and plays an important role in scavenging ROS levels (*De Franceschi et al., 2011*) through

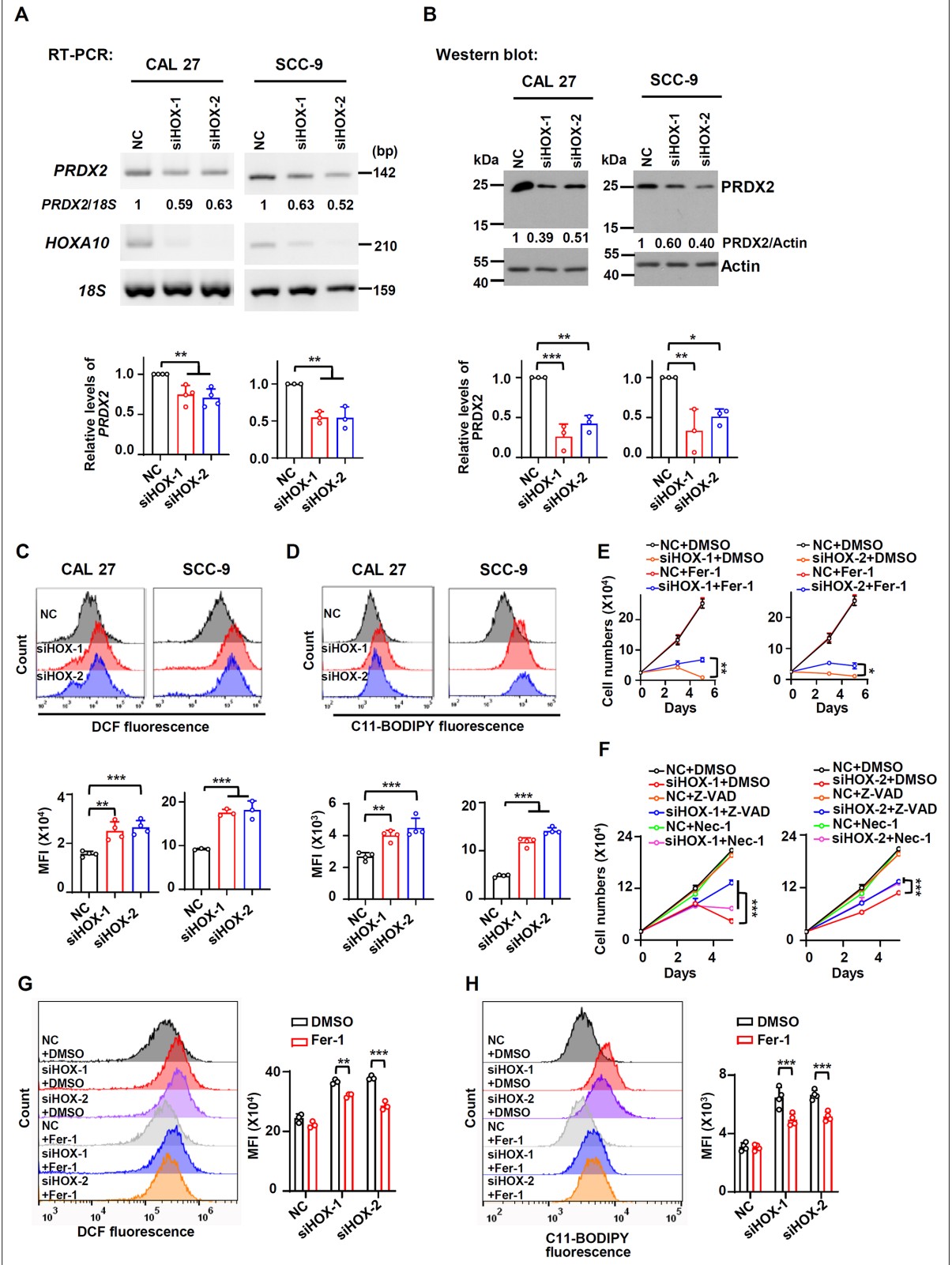

**Figure 6.** HOXA10 promotes the expression of PRDX2 and inhibits ferroptosis in oral squamous cell carcinoma (OSCC). (**A, B**) Effects of HOXA10 knockdown (siHOX-1 and siHOX-2) on PRDX2 expression levels were analyzed by RT-PCR (**A**) or western blot (**B**) in CAL 27 or SCC-9. *18S* rRNA (**A**) or actin (**B**) served as loading controls. Data are means ± SD, n = 4 or 3 for CAL 27, n = 3 for SCC-9. (**C, D**) Effects of HOXA10 knockdown on ROS levels (**C**) or lipid peroxidation (**D**) were detected with DCFH-DA (**C**) or BODIPY 581/591 reagent (**D**) by flow cytometry in CAL 27 or SCC-9. The histograms

*Figure 6 continued on next page*

*Figure 6 continued*

below summarized the levels of MFI. Data are means ± SD, n = 4 for CAL 27, n = 3 or 4 for SCC-9. (**E**) CAL 27 cells transfected with anti-HOXA10 siRNAs were also treated with Fer-1 (10 μM), a ferroptosis inhibitor. Negative control siRNA and DMSO were used as controls. Cells were divided into six groups: NC + DMSO, siHOX-1 + DMSO, siHOX-2 + DMSO, NC + Fer-1, siHOX-1 + Fer-1, and siHOX-2 + Fer-1. To display clear diagrams, the proliferation curves of siHOX-1-treated cells or siHOX-2-treated cells were shown separately. Data are means ± SD, n = 3. (**F**) CAL 27 cells transfected with anti-HOXA10 siRNAs were also treated with Z-VAD (20 μM, an apoptosis inhibitor) or Nec-1 (20 μM, a necroptosis inhibitor). Negative control siRNA and DMSO were used as controls. Cells were divided into nine groups: NC + DMSO, siHOX-1 + DMSO, siHOX-2 + DMSO, NC + Z-VAD, siHOX-1 + Z-VAD, siHOX-2 + Z-VAD, NC + Nec-1, siHOX-1 + Nec-1, and siHOX-2 + Nec-1. To display clear diagrams, the proliferation curves of siHOX-1-transfected cells or siHOX-2-transfected cells were shown separately. Data are means ± SD, n = 3. (**G, H**) The total ROS levels (**G**) or lipid peroxidation (**H**) of cells simultaneously transfected with anti-HOXA10 siRNAs or NC and treated with Fer-1 or DMSO were detected with DCFH-DA (**G**) or BODIPY 581/591 reagent (**H**) by flow cytometry. The histogram on the right summarized the levels of mean fluorescent intensity (MFI). Data are means ± SD, n = 3 for (**G**), n = 4 for (**H**). *p<0.05, **p<0.01, ***p<0.001.

The online version of this article includes the following source data and figure supplement(s) for figure 6:

**Source data 1.** Raw data files for *Figure 6A-D* and *Figure 6G-H*.

**Figure supplement 1.** HOXA10-knockdown inhibits GPX4 expression.

**Figure supplement 1—source data 1.** Raw data files for *Figure 6—figure supplement 1A and B*.

**Figure supplement 2.** Validation of HOXA10 knockdown in *Figure 6*.

**Figure supplement 2—source data 1.** Raw data files for *Figure 6—figure supplement 2A and B*.

**Figure supplement 3.** The increased apoptosis rates caused by HOXA10 knockdown could be rescued by apoptosis inhibitor Z-VAD.

**Figure supplement 3—source data 1.** Raw data file for *Figure 6—figure supplement 3*.

**Figure supplement 4.** The combination of Fer-1 and Z-VAD partially rescues the deduced cell proliferation caused by HOXA10 knockdown.

**Figure supplement 4—source data 1.** Raw data file for *Figure 6—figure supplement 4*.

consuming $H_2O_2$ (*Lu et al., 2014*). We found that isoform II overexpression or knockdown promoted or suppressed PRDX2 expression, respectively. We found that isoform II knockdown also could induce apoptosis in OSCC cells, and PRDX2 overexpression could also reduce the elevated apoptosis induced by isoform II-knockdown. Moreover, isoform II could specifically bind to the promoter of *PRDX2* and then transactivate the PRDX2 expression, thereby inhibiting OSCC cell ferroptosis.

In this study, we also found that RUNX2 isoform II was overexpressed in OSCC tissues and was associated with tumor progression and poor prognosis. In the past, it has been reported that RUNX2 was overexpressed in tumors (*Akech et al., 2010*; *Guo et al., 2021*; *Hong et al., 2020*). Chang et al. demonstrated that RUNX2 was overexpressed in HNSCC samples (*Chang et al., 2017*) and could serve as a poor prognostic marker in HNSCC (*Chang et al., 2016*). However, only a few articles have addressed the expression levels of isoforms. For example, isoform I is the major variant in papillary thyroid carcinomas (*Sancisi et al., 2012*), and isoform II is highly expressed in non-small cell lung cancer (*Herreño et al., 2019*). However, the expression levels of isoforms in OSCC are unknown. Surprisingly, our results showed that there was no statistically significant difference in expression levels of total RUNX2 between OSCC patients and normal controls from the TCGA database. Interestingly, we discovered that the expression levels of isoform I were lower in TCGA OSCC patients than in normal tissues, while the expression levels of isoform II were overexpressed in TCGA OSCC tissues. Furthermore, we proved with clinical samples that the relative expression levels of isoform II were higher in OSCC tissues than those in normal tissues.

Next, we uncovered the underlying mechanisms of RUNX2 isoform II to improve proliferation in OSCC cell lines. In the past, it has been reported that overexpression or knockdown of total RUNX2 in HNSCC cell lines could promote or inhibit cell proliferation, respectively (*Chang et al., 2017*). In this study, we found that isoform II overexpression or specific isoform II-knockdown could promote or suppress OSCC cell proliferation, respectively. In addition, we also demonstrated that isoform II was required for in vivo OSCC tumorigenesis.

Subsequently, we discovered that HOXA10 knockdown inhibited the expression of *RUNX2* isoform II and led to ferroptosis and apoptosis. HOXA10, a transcription factor, plays an important role in tumor progression (*Li et al., 2022a*; *Song et al., 2019*). HOXA10 was reported to transactivate *Runx2* P1 promoter (*Hassan et al., 2007*). Similarly, we found that the relative expression levels of HOXA10 were positively associated with the expression levels of isoform II in OSCC patients from TCGA and HOXA10 knockdown led to the downregulation of isoform II in OSCC cells. It has been demonstrated

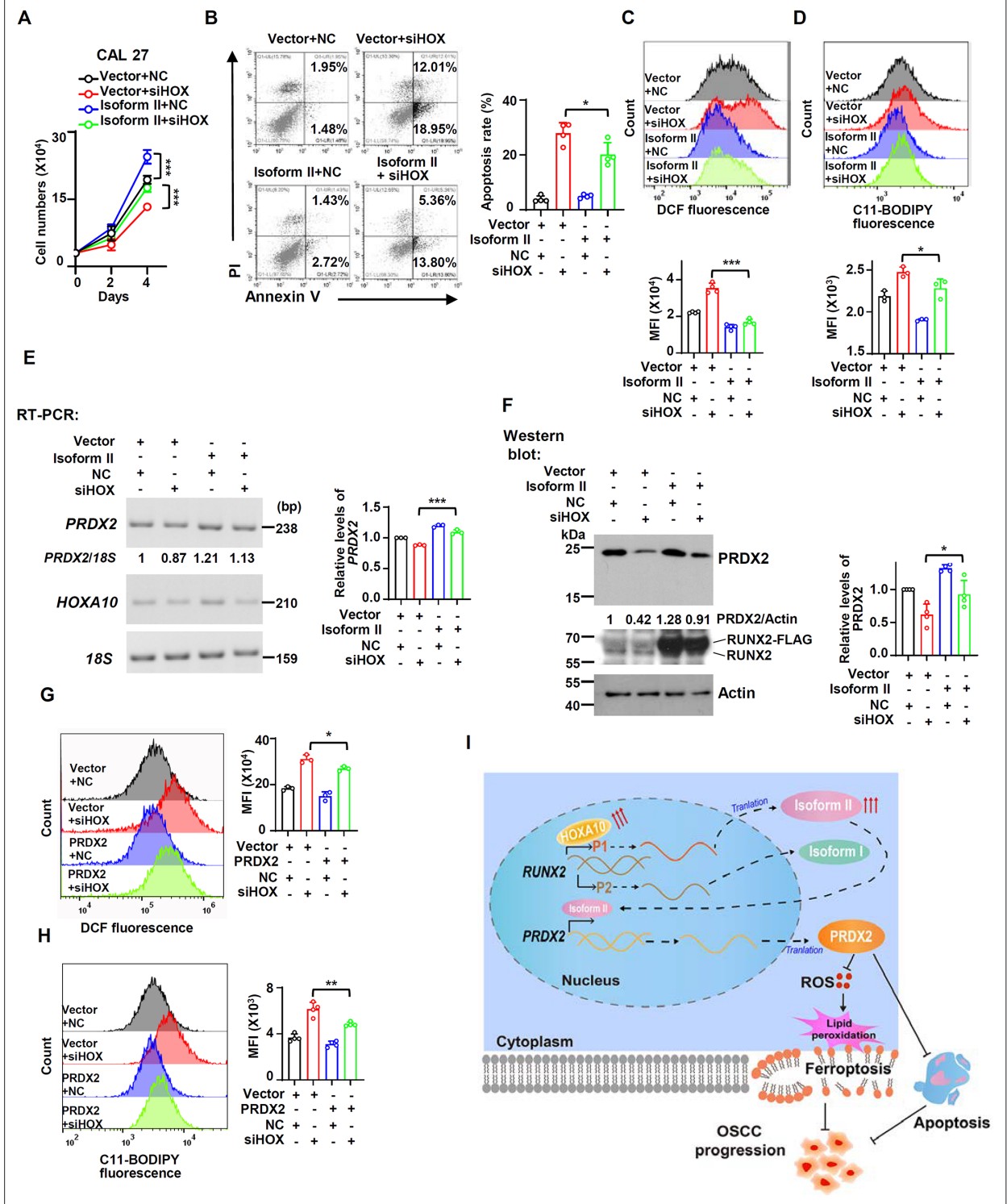

**Figure 7.** Ferroptosis and apoptosis induced by HOXA10-knockdown can be rescued by isoform II overexpression or PRDX2 overexpression. (A–F) CAL 27 cells were co-transfected with isoform II-expression lentivirus, empty control lentivirus, and HOXA10 siRNA (siHOX), negative control siRNA (NC). Transfected cells were divided into four groups: Vector + NC, Vector + siHOX, Isoform II + NC, and Isoform II + siHOX. (A) Cell number was counted on days 2 and 4. Data are means ± SD, n = 5. (B) The cellular apoptosis of transfected cells was analyzed by flow cytometry. The histogram on the right summarized the cell apoptosis. Data are means ± SD, n = 4. (C, D) The total reactive oxygen species (ROS) levels (C) or lipid peroxidation (D) of transfected cells were detected with DCFH-DA (C) or BODIPY 581/591 reagent (D) by flow cytometry. The histograms below summarized the levels of mean fluorescent intensity (MFI). Data are means ± SD, n = 4 or 3. (E, F) Effect of HOXA10-knockdown in isoform II-overexpressed cells on PRDX2

*Figure 7 continued on next page*

*Figure 7 continued*

expression levels was analyzed by RT-PCR (**E**) or western blot (**F**). *18S* rRNA (**E**) or actin (**F**) served as loading controls. Data are means ± SD, n = 3 or 4. (**G, H**) CAL 27 cells were co-transfected with PRDX2-expression lentivirus, empty control lentivirus, and siHOX, NC. Transfected cells were divided into four groups: Vector + NC, Vector + siHOX, PRDX2 + NC, and PRDX2 + siHOX. The total ROS levels (**G**) or lipid peroxidation (**H**) of transfected cells were detected with DCFH-DA (**G**) or BODIPY 581/591 reagent (**H**) by flow cytometry. The histograms on the right summarized the levels of MFI. Data are means ± SD, n = 3. (**I**) The model of a new ferroptosis-related or apoptosis-related pathway-HOXA10/RUNX2 isoform II/PRDX2 in this study. *p<0.05, **p<0.01, ***p<0.001.

The online version of this article includes the following source data and figure supplement(s) for figure 7:

**Source data 1.** Raw data files for *Figure 7B-H*.

**Figure supplement 1.** Validation of expression levels of HOXA10, RUNX2 isoform II, and PRDX2 in *Figure 7*.

**Figure supplement 1—source data 1.** Raw data files for *Figure 7—figure supplement 1A-D*.

**Figure supplement 2.** PRDX2 overexpression rescues the apoptosis induced by isoform II knockdown.

**Figure supplement 2—source data 1.** Raw data file for *Figure 7—figure supplement 2*.

that HOXA10 knockdown suppressed cell proliferation and enhanced apoptosis in Fadu cells (*Guo et al., 2018*). We found that HOXA10 knockdown inhibited cell proliferation and promoted cellular apoptosis in CAL 27 and SCC-9 through suppressing RUNX2 isoform II expression. Interestingly, we discovered that HOXA10 knockdown induced ferroptosis through suppressing RUNX2 isoform II and PRDX2 expression, which suggested a new regulatory pathway of anti-ferroptosis.

In conclusion, we identified RUNX2 isoform II as a novel ferroptosis and apoptosis inhibitor in OSCC cells by transactivating PRDX2 expression. RUNX2 isoform II plays oncogenic roles in OSCC. Moreover, we also found that HOXA10 is an upstream regulator of RUNX2 isoform II and is required for suppressing ferroptosis and apoptosis through RUNX2 isoform II and PRDX2. Our results suggest a new regulatory mechanism of ferroptosis through HOXA10/RUNX2 isoform II/PRDX2 pathway (*Figure 7I*) and may provide the novel diagnostic markers and therapeutic targets for OSCC.

## Materials and methods

### Human tissues

The human OSCC tissues and adjacent normal tissues were collected from the Hospital of Stomatology in Wuhan University. The research was approved by the Ethics Committee at the Hospital of Stomatology in Wuhan University (2023-B03), and the study methodologies conformed with standards of the Declaration of Helsinki. Informed consents were obtained from all participants. Eleven patients diagnosed with OSCC were used in this study. All histologic diagnoses were performed by the Department of Pathology of the Hospital of Stomatology. The characteristics of the 11 OSCC patients are summarized in *Supplementary file 1*.

### Nude mice xenograft tumor formation assay

BALB/c nude mice (female, 5–6 weeks) were purchased from Beijing Vital River Laboratory Animal Technology Co, Ltd (Vital River, Beijing, China). Animal experiments comply with the ARRIVE guidelines and were performed with the approval of the institutional Animal Ethics Committee, Hospital of Stomatology, Wuhan University (S07922110B).

CAL 27 cells stably transfected with short hairpin RNAs (shRNAs) against RUNX2 isoform II (shisoform II-1, shisoform II-2), with shRNA against PRDX2 (shPRDX2-1, shPRDX2-2) or nonspecific shRNA (shNC) through lentivirus were injected subcutaneously into both sides of the dorsum of nude mice ($3.5 \times 10^5$ cells per injection side, six mice per group). Tumor sizes were monitored every 2 or 3 days. Tumor volume was calculated as Length × Width$^2$ ×0.52. Tumor weights were acquired after the mice were sacrificed.

### Cell culture

CAL 27 and SCC-9 cells were obtained as previously reported (*Yang et al., 2018*) and authenticated using STR profiling analysis. HEK 293T cells were obtained from Procell Life Science (Procell, Wuhan, China) and authenticated using STR profiling analysis. Cell lines routinely tested negative for mycoplasma contamination. CAL 27 and HEK 293T were maintained in Dulbecco's modified Eagle's

medium (DMEM) (Hyclone, Marlborough, MA, USA) supplemented with 10% fetal bovine serum (FBS) (Gibco, Carlsbad, CA, USA) and 1% antibiotic-antimycotic (Gibco). SCC-9 was cultured in DMEM/F12 medium (Hyclone) with 10% FBS, 1% antibiotic-antimycotic and 400 ng/mL hydrocortisone (Sangon Biotech, Shanghai, China). All cells were incubated at 37°C in 5% $CO_2$ humidified air.

## Plasmids and transfection

The gene fragments of human *RUNX2* isoform II and isoform I were amplified from CAL 27 cDNA by using the primers 5′ ATGGCATCAAACAGCCTCTTC 3′ and 5′ ATATGGTCGCCAAACAGATTCATC 3′, 5′ ATGCGTATTCCCGTAGATCCG 3′ and 5′ ATATGGTCGCCAAACAGATTCATC 3′, respectively. Then the PCR products were cloned into p3XFLAG-CMV-14 at BamHI and EcoRV sites. The respective FLAG-fusion fragments were then cloned into pLVX-IRES-puro vector at EcoRI and SpeI sites to obtain the recombinant expression plasmids isoform II or isoform I.

The gene fragment of human *PRDX2* was amplified from CAL 27 cDNA by using the primers 5′ ATCGTCCGTGCGTCTAGCCTT 3′ and 5′ ATTGTGTTTGGAGAAATATTCCTTGCT 3′. The PCR product was reamplified by 5′ TTCCGGAATTCGCCACCATGGCCTCCGGTAA 3′ and 5′ TTCGCGCGGCCG CCTACTTGTCATCGTCATCCTTGTAGTCGATGTCATGATCTTTATAATCACCGTCATGGTCTTTGTAG TCTTTTGCCGCAGCTTC 3′ to obtain PRDX2-FLAG fusing fragment, and then was cloned into pLVX-IRES-puro vector at EcoRI and NotI to obtain the recombinant expression plasmid PRDX2.

HEK-293T cells were co-transfected with 2 µg lentiviral backbone plasmids (isoform II, isoform I, PRDX2 or control vector, pLVX-IRES-puro) and the packaging plasmids psPAX2 and pMD2.G at a ratio 4:3:1 in the presence of Lipofectamine 2000 (Invitrogen, Carlsbad, CA, USA). The supernatants containing lentiviral particles were collected 48 hours after transfection. OSCC cell lines were then transfected with lentiviral supernatants in the presence of polybrene (Santa Cruz, Dallas, TX, USA).

## TCGA datasets

We downloaded the data of total RUNX2 and HOXA10 expression levels, and the clinical data in patients with OSCC of TCGA HNSCC dataset from the online program TSVdb (http://www.tsvdb. com). Then, we reanalyzed the expression levels of total RUNX2 and HOXA10 between OSCC patients and normal controls (309 OSCC cases and 32 normal cases). The percent-splice-in (PSI) value data representing the expression levels of exon 1.1 (isoform II) or exon 2.1 (isoform I) of OSCC (288 OSCC cases and 27 normal cases with PSI values) and other carcinomas were obtained from the online program TCGA SpliceSeq (https://bioinformatics.mdanderson.org/TCGA/SpliceSeq).

## RNA extraction and reverse transcription polymerase chain reaction (RT-PCR)

Total RNA was purified using the AxyPrep multisource total RNA miniprep Kit (Axygen, Union City, CA, USA) according to the manufacturer's protocol. Total RNA was treated with DNase I (Thermo Fisher Scientific, Carlsbad, CA, USA) to remove genomic DNA contamination and then was reversely transcribed by using the Maxima H Minus cDNA Synthesis Master Mix (Thermo Fisher Scientific) in accordance with the manufacturer's protocol. The cDNA was subject to PCR amplification using Green Taq Mix (Vazyme, Nanjing, China) with different primers. The primer sequences are listed in *Supplementary file 2*.

## RNA interference and transfection

The short interfering RNAs (siRNAs) against human RUNX2 isoform II were synthesized by Sangon Biotech. The sequences are as follows: 5′ GCUUCAUUCGCCUCACAAACA 3′ (si-II-1) and 5′ GGUU AAUCUCCGCAGGUCACU 3′ (si-II-2). The siRNAs, synthesized by GenePharma (Suzhou, China), were used to knock down human HOXA10 (siHOX-1 and siHOX-2). The sequences of siHOX-1 and siHOX-2 are 5′ GAGUUUCUGUUCAAUAUGUACCUUA 3′ and 5′ CCGGGAGCUCACAGCCAACUUUAAU 3′, respectively. Cells were transfected with 20 nM siRNAs in the presence of Lipofectamine 3000 (Invitrogen) according to the manufacturer's protocol. After 48 hours, cells were transfected again.

The short hairpin RNAs (shRNAs) against RUNX2 isoform II (shisoform II-1, shisoform II-2) plasmids, against PRDX2 (shPRDX2-1, shPRDX2-2) plasmids and non-specific shRNA (shNC) plasmid were provided by Vector Builder Inc (Vector Builder, Guangzhou, China). The target sequences of shisoform II-1 and shisoform II-2 are GGTTAATCTCCGCAGGTCACT and GCTTCATTCGCCTCACAAACA.

The target sequences of shPRDX2-1 and shPRDX2-2 are GCCTGGCAGTGACACGATTAA and CCTT CGCCAGATCACTGTTAA. The plasmids shisoform II or shNC were co-transfected with psPAX2 and pMD2.G into HEK 293T cells to produce supernatants containing lentiviruses. Then the supernatants containing lentiviral particles were collected 48 hours after transfection to transfect CAL 27 OSCC cell line in the presence of polybrene.

## Immunohistochemistry (IHC)

The mouse tumor tissues or human OSCC tissues were fixed in 4% paraformaldehyde (Servicebio, Wuhan, China), and embedded in paraffin, then sectioned in 4 μm. The staining of PRDX2, TFRC, or 4-HNE was detected by using a Dako EnVision FLEX kit (Dako, Glostrup, Denmark) according to the manufacturer's instructions. Briefly, the sections were subjected to antigen retrieval in Target Retrieval Solution (Dako). Endogenous peroxidase was blocked with Peroxidase-Blocking Reagent (Dako). Then, the sections were incubated with primary antibody (PRDX2, TFRC, or 4-HNE) at 4°C overnight and followed by incubation with secondary antibody FLEX/HRP (Dako) at room temperature for 30 min. Staining was developed by diaminobenzidine (DAB) substrate (Dako). The stained sections were scanned by Pannoramic MIDI (3D HISRECH). The concentration of PRDX2 (Proteintech, Wuhan, China, #10545-2-AP), 4-HNE (Abcam, Cambridge, UK, ab48506), or TFRC (Abcam, ab214039) used in this study was 1:1000, 1:600, or 1:500, respectively.

## Chromatin immunoprecipitation and quantitative PCR (ChIP and qPCR)

ChIP was performed in CAL 27 cells stably transfected by FLAG-tagged RUNX2 isoform II or vector (pLVX-IRES-puro) using ABclonal Sonication Chip Kit (ABclonal, Wuhan, China) according to the manufacturer's protocol. In brief, cells in 10 cm culture dishes were crosslinked with 1% formaldehyde and the reaction was terminated by glycine. Cells were lysed, and samples were then sonicated to disrupt the nuclear membrane. After centrifugation, the supernatants were collected which contained the chromatin. Chromatin solutions were, respectively, incubated with antibodies anti-FLAG (2 μg, Proteintech, #20543-1-AP) and anti-normal rabbit IgG (2 μg, ABclonal, #RM20712). Then, they were rotated at 4°C for 6 hours, followed by incubation with ChIP-grade protein A/G magnetic beads (ABclonal, #RM02915) at 4°C for 2 hours. After washing, the cross-links were reversed at 65°C overnight, and DNA was purified (ABclonal, #RK30100) and then used for ChIP-qPCR analysis. For the ChIP-qPCR experiments with a pair of primers for the *PRDX2* promoter region as follows: 5′ TACA GGTGTGAGCCAGCCACCAT 3′ (forward primer) and 5′ TGGCGGGCACCAAGGATGTTGT 3′ (reverse primer).

## Western blot

Cells were lysed with 2 × SDS sample buffer, and then the total cellular protein was denatured for 3 min at 95°C. Total cellular proteins were separated in 4–12% YoungPAGE Bis-Tris gels (GenScript, Nanjing, China) or 10% gels using One-Step PAGE Gel Fast Preparation Kit (Vazyme), transferred to nitrocellulose membrane (Pall Corporation, USA), followed by the block with 5% (w/v) non-fat milk (Servicebio) for 1 hour. Then the membranes were incubated with mouse RUNX2 antibody (1:500, Santa Cruz, #sc-390351), rabbit FLAG antibody (1:2000, Proteintech, #20543-1-AP), rabbit PRDX2 antibody (1:2000, Proteintech, #10545-2-AP), GPX4 antibody (1:1000, ABclonal, #A11243), and mouse actin antibody (1:5000, Proteintech, #66009-1-Ig).

## Cell counting and colony formation assay

Cell counting was performed by the trypan blue exclusion method using 0.4% trypan blue solution (Biosharp, Hefei, China). 1000 CAL 27 or SCC-9 cells were seeded into 6-well plates and cultured in complete medium for 11 days at 37°C. Then, cells were fixed with 4% paraformaldehyde and stained with crystal violet (Servicebio). The number of colonies (at least 50 cells/colony) was counted.

## Wound healing assay

For wound healing, cells with stable expression of isoform II, isoform I, or vector were plated in six-well plates and grown to 90–100% concentration. The artificial wounds were created with pipette tips, and then cells were cultured in serum-free medium after washing with PBS. The migrating cells in the wound front were imaged at different times. Wound healing status was analyzed by ImageJ.

## Transwell migration and invasion assays

The migration and invasion assays were performed using Transwell chamber (Corning, USA). For migration assay, $5 \times 10^4$ SCC-9 with stable expression of isoform II, isoform I, or vector per well were seeded into the upper chambers with serum-free medium. The lower chambers were contained the complete medium. For invasion assay, the 8 µm pore transwell filters were pre-coated with Matrigel (Corning), and the subsequent steps were similar to the migration assay. After the cells migrated or invaded for certain time, they were fixed with 4% paraformaldehyde, stained by 0.1% crystal violet. The transwell filters were imaged and analyzed using ImageJ.

## Apoptosis assay

Cell apoptosis was analyzed using the Annexin V-FITC/PI apoptosis assay kit (KeyGEN BioTECH, Nanjing, China, #KGA108). Briefly, the cells were collected in 200 µL binding buffer with 2 µL of Annexin V-FITC and 2 µL PI, and incubated for 20 min under dark conditions. Cellular apoptosis was evaluated using flow cytometry.

## ROS detection

The total cellular ROS was detected using a ROS assay kit DCFH-DA (Beyotime, Shanghai, China, #S0033S). DCFH-DA was diluted to a final concentration of 10 µM. Then, OSCC cell lines were collected and suspended in diluted DCFH-DA in the dark at 37°C for 25 min and washed three times with PBS. The samples were analyzed using flow cytometry.

## Transmission electron microscopy (TEM)

Cells cultured in a 6-well plate were collected and fixed with a solution containing 2.5% glutaral-dehyde (Servicebio) for 2 hours in the dark at room temperature. Then the ultrathin sections were made by Servicebio and visualized by using the JEM-1011 transmission electron microscope (Hitachi, Japan). The length of mitochondria was analyzed by Nano Measurer, and the average mitochondrial length of each group was summarized.

## Detection of lipid peroxidation

OSCC cells were collected and suspended in PBS containing 5 µM C11-BODIPY 581/591 (Thermo Fisher Scientific, #D3861) in the dark at 37°C for 30 min and washed three times with PBS, and the samples were analyzed using flow cytometry through the FITC channel.

## Oxygen consumption rate (OCR) assay

To assess the impact of isoform II knockdown on mitochondrial respiration in OSCC cells, Seahorse Bioscience XFe24 (Agilent Seahorse XFe24 Analyzer) was used according to the manufacturer's instructions. The mitochondrial respiration was analyzed using the Mito-stress Test Kit. Briefly, $3 \times 10^4$ SCC-9 cells were seeded in 24-well plates and incubated overnight. Cells were washed and incubated in Seahorse detection buffer at 37°C in no $CO_2$ for 1 hour. The OCR was tested using the compounds of oligomycin (1 µM), FCCP (1 µM) and rotenone and antimycin A (Rot/AA, 1 µM).

## Half-maximal inhibitory concentration (IC$_{50}$) of RSL3

Cells were seeded in 24-well plates ($2 \times 10^4$ cells/well) overnight, then treated with RSL3 at different concentration (0.125, 0.25, 0.5, 1, 2, 4, 8, 16, 32 µM). The cells were counted on the next day. The IC$_{50}$ values of RSL3 were analyzed using GraphPad Prism.

## Statistical analysis

The comparison of the mean values between three groups or more was performed using the one-way ANOVA test in the GraphPad Prism software. The Mann–Whitney test was used to compare the mean differences of RUNX2 exon 1.1 (isoform II) or exon 2.1 (isoform I) in COAD and PRAD from TCGA, the mean difference of isoform II expression levels (isoform II/*GAPDH*) or *HOXA10* expression levels in our clinical samples, the weight and volume of mouse tumors (shisoform II), the H score of 4-HNE staining of mouse tumors, and the expression levels of HOXA10 in TCGA OSCC patients. All remaining two-group comparisons of means were analyzed by Student's *t*-test. The Kruskal–Wallis test was used to compare the tumor growth in nude mice between nonspecific control and shPRDX2. Survival analysis

was performed with a log-rank test, and survival curve was produced using the Kaplan–Meier method in the GraphPad Prism software. The correlation of RUNX2 exon 1.1 (isoform II) and HOXA10 was calculated with the Spearman rank method. The quantification of RT-PCR was realized using ImageJ software. p<0.05 was considered statistically significant.

## Key resources table

| Reagent type (species) or resource | Designation | Source or reference | Identifiers | Additional information |
|---|---|---|---|---|
| Strain, strain background (*Mus musculus*) | BALB/c nude mice | Vital River | Strain No.401 | |
| Cell line (human) | CAL 27 SCC-9 | *Yang et al., 2018* (PMID:29857020) | | |
| Cell line (human) | HEK 293T | Procell | Cat#CL-0005 | |
| Transfected construct (human) | RUNX2 isoform II shRNAs (shisoform II-1 and shisoform II-2) | Vector Builder | Cat#VB221206-1024udv; VB221206-1021ujk | Lentiviral construct to transfect and express the shRNA |
| Transfected construct (human) | PRDX2 shRNAs (shPRDX-1 and shPRDX2-2) | Vector Builder | Cat#VB900064-6571eqq; VB900064-6578nnu | Lentiviral construct to transfect and express the shRNA |
| Transfected construct (human) | Negative control shRNA (shNC) | Vector Builder | Cat#VB010000-009mxc | |
| Transfected construct (human) | siRNAs to isoform II (si-II-1/si-II-2) | Sangon Biotech | | Transfected construct (Human) |
| Transfected construct (human) | siRNAs to HOXA10 (siHOX-1/ siHOX-2) | GenePharma | | Transfected construct (Human) |
| Transfected construct (human) | siRNA to negative control (siNC) | Sangon Biotech | | |
| Biological sample (human) | Oral squamous cell carcinoma | Hospital of Stomatology, Wuhan University | | The adjacent normal tissues were also acquired |
| Antibody | Anti-human RUNX2 (mouse monoclonal) | Santa Cruz | Cat#SC-390351; RRID:AB_2892645 | WB (1:500) |
| Antibody | Anti-4-HNE (mouse monoclonal) | Abcam | Cat#ab48506; RRID:AB_867452 | IHC (1:600) |
| Antibody | Anti-TFRC (rabbit monoclonal) | Abcam | Cat#ab214039; RRID:AB_2904534 | WB (1:2000) IHC (1:500) |
| Antibody | Anti-human FLAG (rabbit polyclonal) | Proteintech | Cat#20543-1-AP; RRID:AB_11232216 | WB (1:2000) ChIP (2 µg) |
| Antibody | Anti-human/mouse PRDX2 (rabbit polyclonal) | Proteintech | Cat#10545-2-AP; RRID:AB_2168202 | WB (1:2000) IHC (1:1000) |
| Antibody | Anti-human actin (mouse monoclonal) | Proteintech | Cat#66009-1-lg; RRID:AB_2782959 | WB (1:5000) |
| Antibody | Anti-GPX4 (rabbit monoclonal) | ABclonal | Cat#A11243 RRID:AB_2861533 | WB (1:1000) |
| Recombinant DNA reagent | RUNX2 isoform II (plasmid) | This paper | | FLAG-tagged isoform II |
| Recombinant DNA reagent | RUNX2 isoform I (plasmid) | This paper | | FLAG-tagged isoform I |
| Recombinant DNA reagent | PRDX2 (plasmid) | This paper | | FLAG-tagged PRDX2 |
| Commercial assay or kit | Total RNA Miniprep Kit | Axygen | Cat#AP-MN-MS-RNA-250 | |
| Commercial assay or kit | Green Taq Mix | Vazyme | Cat#P131-AA | |
| Commercial assay or kit | Phanta Super-Fidelity DNA Polymerase | Vazyme | Cat#P505-d1 | |

*Continued on next page*

*Continued*

| Reagent type (species) or resource | Designation | Source or reference | Identifiers | Additional information |
|---|---|---|---|---|
| Commercial assay or kit | Maxima H Minus cDNA Synthesis Master Mix | Thermo Fisher Scientific | Cat#M1682 | |
| Commercial assay or kit | ChamQ Universal SYBR qPCR Master Mix | Vazyme | Cat#Q711-02 | |
| commercial assay or kit | Reactive Oxygen Species Assay Kit | Beyotime | Cat#S0033S | |
| Commercial assay or kit | BODIPY 581/591C11 | Thermo Fisher Scientific | Cat#D3861 | |
| Commercial assay or kit | Annexin V-FITC/PI apoptosis assay kit | KeyGEN BioTECH | Cat#KGA108 | |
| Commercial assay or kit | Sonication ChIP Kit | Abclonal | Cat#RK20258 | |
| Commercial assay or kit | Protein A/G beads | Abclonal | Cat#RM02915 | |
| Commercial assay or kit | Protease Inhibitor Cocktail | Abclonal | Cat#RM02916 | |
| Commercial assay or kit | EnVision FLEX TARGET RETRIEVAL SOLUTION HIGH pH | Dako | REF#K8023 | |
| Commercial assay or kit | EnVision FLEX PEROXIDASE-BLOCOING REAGENT | Dako | REF#GV800 | |
| Commercial assay or kit | EnVision FLEX/HRP | Dako | REF#K8023 | |
| Commercial assay or kit | Liquid DAB+ Substrate Chromogen System | Dako | REF#K3468 | |
| Commercial assay or kit | XF Cell Mito Stress Test Kit | Agilent | Cat#103015-100 | |
| Commercial assay or kit | Seahorse XF DMEM Assay Medium Pack | Agilent | Cat#103680-100 | |
| Commercial assay or kit | Seahorse XFe24 FluxPak | Agilent | Cat#102340wz-100 | |
| Chemical compound, drug | Ferrostatin-1 | MCE | CAS#347174-05-4 | Dissolved in DMSO |
| Chemical compound, drug | RSL3 | MCE | CAS#1219810-16-8 | Dissolved in DMSO |
| Chemical compound, drug | Z-VAD | Selleck | Cat#S7023 | Dissolved in DMSO |
| Chemical compound, drug | Necrostatin-1 | Selleck | Cat#S8037 | Dissolved in DMSO |
| Chemical compound, drug | Polybrene | Santa Cruz | Cat#sc-134120 | |
| Chemical compound, drug | Lipofectamine 2000 | Thermo Fisher Scientific | Cat#11668019 | |
| Chemical compound, drug | Lipofectamine 3000 | Thermo Fisher Scientific | Cat#L3000001 | |
| Software, algorithm | FlowJo | FlowJo | RRID:SCR_008520 | |
| Software, algorithm | GraphPad Prism | GraphPad Software | RRID:SCR_002798 | |
| Software, algorithm | CytExpert | Beckman Coulter | RRID:SCR_017217 | |
| Software, algorithm | ImageJ | National Institutes of Health | RRID:SCR_003070 | |

## Materials availability

All materials may be made available to the scientific community upon request.

## Acknowledgements

This study was supported by the National Natural Science Foundation of China grant numbers 81970933 and 82170966.

## Additional information

### Funding

| Funder | Grant reference number | Author |
|---|---|---|
| National Natural Science Foundation of China | 81970933 | Jihua Guo |
| National Natural Science Foundation of China | 82170966 | Rong Jia |

The funders had no role in study design, data collection and interpretation, or the decision to submit the work for publication.

### Author contributions

Junjun Huang, Data curation, Formal analysis, Validation, Investigation, Visualization, Methodology, Writing - original draft; Rong Jia, Jihua Guo, Conceptualization, Resources, Supervision, Funding acquisition, Validation, Methodology, Project administration, Writing – review and editing

### Author ORCIDs

Rong Jia https://orcid.org/0000-0001-6960-6199
Jihua Guo https://orcid.org/0000-0002-6713-4439

### Ethics

The human OSCC tissues and adjacent normal tissues were collected from the Hospital of Stomatology in Wuhan University. The research was approved by the Ethics Committee at the Hospital of Stomatology in Wuhan University (2023-B03) and the study methodologies conformed with standards of the Declaration of Helsinki. Informed consents were obtained from all participants. Eleven patients diagnosed with OSCC were used in this study. All histologic diagnoses were performed by the Department of Pathology of the Hospital of Stomatology. The characteristics of the eleven OSCC patients were summarized in Supplementary File 1.

BALB/c nude mice (female, 5-6 weeks) were purchased from Beijing Vital River Laboratory Animal Technology Co., Ltd (Vital River, Beijing, China). Animal experiments comply with the ARRIVE guidelines and were performed with the approval of the institutional Animal Ethics Committee, Hospital of Stomatology, Wuhan University (S07922110B).

Reviewer #1 (Public review): https://doi.org/10.7554/eLife.99122.3.sa1
Author response https://doi.org/10.7554/eLife.99122.3.sa2

## Additional files

### Supplementary files

Supplementary file 1. The clinical characteristics of OSCC patients.
Supplementary file 2. Primer sequences for RT-PCR.
MDAR checklist

### Data availability

All data generated or analysed during this study are included in the manuscript and supporting files. The original gels/blots generated in this study have been provided in the source data files.

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
