## [Editor Report · eLife Assessment]

This article investigates how isoform II of transcription factor RUNX2 promotes cell survival and proliferation in oral squamous cell carcinoma cell lines. The authors used gain and loss of function techniques to provide **convincing** evidence showing that RUNX2 isoform silencing led to cell death via several mechanisms including apoptosis and ferroptosis that was partially suppressed through RUNX2 regulation of PRDX2 expression. The study provides **valuable** insight into the underlying mechanism by which RUNX2 acts in oral squamous cell carcinoma.

---

## [Referee Report · Reviewer #1 (Public review)]

Summary:

In this paper, authors investigated the role of RUNT-related transcription factor 2 (RUNX2) in oral squamous carcinoma (OSCC) growth and resistance to ferroptosis. They found that RUNX2 suppresses ferroptosis through transcriptional regulation of peroxiredoxin-2. They further explored the upstream positive regulator of RUNX2, HOXA10 and found that HOXA10/RUNX2/PRDX2 axis protects OSCC from ferroptosis.

Strengths:

The study is well designed and provides a novel mechanism of HOXA10/RUNX2/PRDX2 control of ferroptosis in OSCC.

Weaknesses:

According to the data presented in (Figure 2F, Figure 3F and G, Figure 5D and Figure 6E and F), apoptosis seems to be affected in the same amount as ferroptosis by HOXA10/RUNX2/PRDX2 axis, which raises a question on the authors' specific focus on ferroptosis in this study. Reasonably, authors should adapt the title and the abstract in a way that it recapitulates the whole data, which is HOXA10/RUNX2/PRDX2 axis control of cell death, including ferroptosis and apoptosis in OSCC.

Comments on revisions:

The revised manuscript has been well improved, and I'm satisfied with the authors' response to my comments.

---

## [Author Response]

The following is the authors’ response to the original reviews

**eLife Assessment**
This paper investigates how isoform II of transcription factor RUNX2 promotes cell survival and proliferation in oral squamous cell carcinoma cell lines. The authors used gain and loss of function techniques to provide incomplete evidence showing that RUNX2 isoform silencing led to cell death via several mechanisms including ferroptosis that was partially suppressed through RUNX2 regulation of PRDX2 expression. The study provides useful insight into the underlying mechanism by which RUNX2 acts in oral squamous cell carcinoma, but the conclusions of the authors should be revised to acknowledge that ferroptosis is not the only cause of cell death.

We appreciate the editor’s positive comments on our work and the valuable suggestions provided by the reviewers. We did find that RUNX2 isoform II knockdown or HOXA10 knockdown could also lead to apoptosis. We have revised our title as following: “RUNX2 Isoform II Protects Cancer Cells from Ferroptosis and Apoptosis by Promoting PRDX2 Expression in Oral Squamous Cell Carcinoma”. In addition, we have also revised our conclusions in the abstract as follows: “OSCC cancer cells can up-regulate RUNX2 isoform II to inhibit ferroptosis and apoptosis, and facilitate tumorigenesis through the novel HOXA10/RUNX2 isoform II/PRDX2 pathway.” We have added more experiments to better support our conclusions. Please see following responses to reviewers.

**Public Reviews:**

**Reviewer #1 (Public Review):**
Summary:In this paper, authors investigated the role of RUNT-related transcription factor 2 (RUNX2) in oral squamous carcinoma (OSCC) growth and resistance to ferroptosis. They found that RUNX2 suppresses ferroptosis through transcriptional regulation of peroxiredoxin-2. They further explored the upstream positive regulator of RUNX2, HOXA10 and found that HOXA10/RUNX2/PRDX2 axis protects OSCC from ferroptosis.Strengths:The study is well designed and provides a novel mechanism of HOXA10/RUNX2/PRDX2 control of ferroptosis in OSCC.Weaknesses:According to the data presented in (Figure 2F, Figure 3F and G, Figure 5D and Figure 6E and F), apoptosis seems to be affected in the same amount as ferroptosis by HOXA10/RUNX2/PRDX2 axis, which raises questions on the authors' specific focus on ferroptosis in this study. Reasonably, authors should adapt the title and the abstract in a way that recapitulates the whole data, which is HOXA10/RUNX2/PRDX2 axis control of cell death, including ferroptosis and apoptosis in OSCC.

We really grateful for your comments. We agree that these figures do show that isoform II-knockdown or HOXA10-knockdown could induce apoptosis. We have adapted the title and abstract as follow:

Title: “RUNX2 Isoform II Protects Cancer Cells from Ferroptosis and Apoptosis by Promoting PRDX2 Expression in Oral Squamous Cell Carcinoma”.

Abstract: “In the present study, we surprisingly find that RUNX2 isoform II is a novel ferroptosis and apoptosis suppressor. RUNX2 isoform II can bind to the promoter of peroxiredoxin-2 (PRDX2), a ferroptosis inhibitor, and activate its expression. Knockdown of RUNX2 isoform II suppresses cell proliferation in vitro and tumorigenesis in vivo in oral squamous cell carcinoma (OSCC). Interestingly, homeobox A10 (HOXA10), an upstream positive regulator of RUNX2 isoform II, is required for the inhibition of ferroptosis and apoptosis through the RUNX2 isoform II/PRDX2 pathway. Consistently, RUNX2 isoform II is overexpressed in OSCC, and associated with OSCC progression and poor prognosis. Collectively, OSCC cancer cells can up-regulate RUNX2 isoform II to inhibit ferroptosis and apoptosis, and facilitate tumorigenesis through the novel HOXA10/RUNX2 isoform II/PRDX2 pathway.”

In addition, we have performed the rescue experiment showing that PRDX2 overexpression rescues the apoptosis induced by isoform II-knockdown (Figure 4-figure supplement 4) or HOXA10-knockdown (Figure 7-figure supplement 2).

We have added the description about these experiments in result “RUNX2 isoform II promotes the expression of PRDX2” and “HOXA10 inhibits ferroptosis and apoptosis through RUNX2 isoform II” as follow: “In addition, we found that PRDX2 overexpression could partially reduce the increased apoptosis caused by isoform II-knockdown. (Figure 4-figure supplement 4).” “PRDX2 overexpression also could rescue the increased cellular apoptosis caused by HOXA10 knockdown (Figure 7-figure supplement 2).”.

Comments:In the description of the result section related to Figure 3E, the author wrote "In addition, we found that isoform II-knockdown induced shrunken mitochondria with vanished cristae with transmission electron microscopy (Figure 3E). These results suggest that RUNX2 isoform II may suppress ferroptosis." The interpretation provided here is not clear to the reviewer. How shrunken mitochondria and vanished cristae can be linked to ferroptosis?

We apologize for the inaccurate description. Ferroptotic cells usually exhibit shrunken mitochondria, reduced or absent cristae, and increased membrane dentistry (Dixon et al., 2012). However, the presence of shrunken mitochondria or vanished cristae does not guarantee that ferroptosis has occurred in the cells. Other evidences, such as the increased ROS production and lipid peroxidation accumulation in cells with RUNX2 isoform II-knockdown must be evaluated as we are showing in Figure 3A and 3B. Furthermore, isoform II overexpression suppressed ROS production (Figure 3C) and lipid peroxidation (Figure 3D). We have revised our interpretation as follow: “In addition, we found that isoform II-knockdown induced shrunken mitochondria with vanished cristae with transmission electron microscopy (Figure 3E). This phenomenon along with the above results of ROS production and lipid peroxidation accumulation assays suggests that RUNX2 isoform II may suppress ferroptosis.”.

Dixon, S. J., Lemberg, K. M., Lamprecht, M. R., Skouta, R., Zaitsev, E. M., Gleason, C. E., . . . Stockwell, B. R. (2012). Ferroptosis: an iron-dependent form of nonapoptotic cell death. Cell, 149(5), 1060-1072. doi:10.1016/j.cell.2012.03.042 PMID:22632970

The electron microscopy images show more elongated mitochondria in the RUNX2 isoform II-KO cells than in RUNX2 isoform II positive cells, which might result from the fusion of mitochondria. These images should complete with a fluorescent mitochondria staining of these cells.

We do find that the TEM images of RUNX2 isoform II-knockdown cells show more elongated mitochondria. The mitochondria undergo cycles of fission and fusion, known as mitochondrial dynamics, which in turn leads to changes in mitochondrial length. Through examining factors related to mitochondrial dynamics, we find that isoform II knockdown could decrease the expression levels of FIS1 (Fission, Mitochondrial 1) (Figure 3-figure supplement 2B) which mediates the fission of mitochondria. Therefore, we speculate that the elongated mitochondria in the isoform II-knockdown cells may be due to the decrease in mitochondrial fission through inhibiting FIS1 expression.

In addition, we have tried our best to perform the fluorescent staining of mitochondrial to observe mitochondrial morphology. However, due to the quality of probes and fluorescent microscope, our images of mitochondrial fluorescence were not satisfactory. So, we re-capture more electron microscopy images, measure the length of mitochondria, and perform statistical analyses. We find that isoform II-knockdown cells show significantly more mitochondrial elongation than the control cells (Author response image 1 and Figure 3-figure supplement 2A). Therefore, we believe that isoform II knockdown promotes mitochondrial elongation to be relatively reliable.

**Author response image 1. sa2fig1:** The new electron microscopy images in RUNX2 isoform II-knockdown cells. RSL3 (a ferroptosis activator) served as a positive control. Scale bar: 1 μm. The calculation and statistical analysis of mitochondrial elongation were added in Figure 3-figure supplement 2A.

What is the oxygen consumption rate in RUNX2 KO cells?

We have performed a new mitochondrial stress assay to analyze the oxygen consumption rate (OCR). We find that RUNX2 isoform II-knockdown can decrease OCR in OSCC cell line. This result has been added to Figure 3-figure supplement 3A and B. It is consistent with our observation of the damaged mitochondria morphology in the cells with RUNX2 isoform II knockdown.

The increase in cell proliferation after RUNX2 overexpression in Figure 2A is not convincing, is there any differences in their migration or invasion capacity?

We agree that overexpression of isoform II didn’t dramatically enhance OSCC cell proliferation. We consider that it may be due to the existing high level of isoform II in OSCC cells. We have performed wound-healing assay and transwell assay to analyze the migration or invasion capacity of cells with RUNX2 isoform II or isoform I overexpression. We find that isoform II overexpression has no effect on the migration and invasion in OSCC cells (Figure 2-figure supplement 2). This phenomenon suggests that further increasing isoform II cannot improve the migration or invasion capacity of OSCC cells. However, isoform I overexpression suppresses the migration and invasion of cancer cells (Figure 2-figure supplement 2), indicating that the upregulation of isoform I, which is downregulated in OSCC cells, may inhibit tumorigenesis. In addition, we found that the expression level of isoform I was lower in TCGA OSCC patients than that in normal controls (Figure 1D), and patients with higher isoform I showed longer overall survival (Figure 1-figure supplement 1). These results support that isoform I may inhibit tumorigenesis in OSCC cells.

The in vivo study shows 50% reduction in primary tumor growth after RUNX2 inhibition by shRNA in CAL 27 xenografts, but only one shRNA is shown. Is this one shRNA clone? At least 2 shRNA clones should be used.

In this vivo primary tumor growth experiment, we used a CAL 27 stable cell line transfected with an shRNA against RUNX2 isoform II (shisoform II-1). We agree that at least two shRNAs should be used. In this revision, we perform another tumor growth experiment with the CAL 27 stably transfected with another new shRNA targeting the different region in isoform II (shisoform II-2). As with the previous experiment, CAL 27 cells stably transfected with this new shRNA also showed significantly reduced tumor growth and weight than those transfected with non-specific control shRNA in nude mice (Figure 2-figure supplement 4A-D).

Apoptosis and necroptosis seem to be affected in the same amount as ferroptosis by HOXA10/RUNX2/PRDX2 axis. This is evident from experiments in Figure 3E, F and from Figure 6E, F and Figure 3G. Either Fer-1, Z-VAD, or Nec-1 used alone, were not able to fully restore cell proliferation to control cell level, which implies an additive effect of ferroptosis, apoptosis and necrosis. The author should verify potential additive or synergistic effect of the combination of Fer-1 and Z-VAD in these assays after si-RUNX2 in Figure 3 F and G and after si-HOX assays.

We sincerely appreciate your valuable comments. We have performed the new assay to analyze the potential additive or synergistic effect of the combination of Fer-1 and Z-VAD after RUNX2 isoform II (si-II) or HOXA10 (si-HOX) knockdown. We find that the combination of Fer-1 and Z-VAD is more effective in rescuing the cell proliferation than Fer-1 or Z-VAD alone. (Figure 3- figure supplement 6 and Figure 6- figure supplement 4).

What is the effect of PRDX2 or HOXA10 depletion on tumor growth?

We have performed a new xenograft tumor formation assay in nude mice to analyze the effect of PRDX2-knockdown on tumor growth. We found that CAL 27 cells stably transfected with shRNAs against PRDX2 showed significantly reduced tumor growth and weight than those transfected with non-specific control shRNA in nude mice (Figure 4-figure supplement 2A-D). Regarding the effect of HOXA10 depletion on tumor growth, please allow us to cite a study (Guo et al., 2018) which demonstrated that HOXA10 knockout in Fadu cells (a cell line of pharyngeal squamous cell carcinoma) could inhibit tumor growth.

We have added these results to the section of “RUNX2 isoform II promotes the expression of PRDX2” as follows: “In line with the inhibitory effect of isoform II-knockdown on tumor growth, CAL 27 cells stably transfected with anti-PRDX2 shRNAs showed notably reduced tumor growth and weight than those transfected with non-specific control shRNA in nude mice (Figure 4-figure supplement 2A-D).”.

Guo, L. M., Ding, G. F., Xu, W., Ge, H., Jiang, Y., Chen, X. J., & Lu, Y. (2018). MiR-135a-5p represses proliferation of HNSCC by targeting HOXA10. Cancer Biol Ther, 19(11), 973-983. doi:10.1080/15384047.2018.1450112 PMID:29580143

What is the clinical relevance of HOXA10 in OSCC patients?

In Figure 5-figure supplement 1B, we have showed that the expression levels of HOXA10 in TCGA OSCC patients were also significantly higher than those in normal controls. In this revision, we further find that patients with higher HOXA10 show significantly shorter overall survival in TCGA OSCC dataset (Figure 5-figure supplement 2C). In addition, we have also analyzed the expression of HOXA10 in our clinical OSCC and adjacent normal tissues, and found that HOXA10 expression level of OSCC tissues is significantly higher than that of normal controls (Figure 5-figure supplement 2A and B), which is consistent with the results from TCGA OSCC dataset.

We have revised our writing in the result “HOXA10 is required for RUNX2 isoform II expression and cell proliferation in OSCC” as follows: “Similarly, HOXA10 expression level of our clinical OSCC tissues is significantly higher than that of adjacent normal tissues (Figure 5-figure supplement 2A and B). Moreover, TCGA OSCC patients with higher expression levels of HOXA10 showed shorter overall survival (Figure 5-figure supplement 2C).”

**Reviewing editor (Public Review):**
This paper reports the role of the Isoform II of RUNX2 in activating PRDX2 expression to suppress ferroptosis in oral squamous cell carcinoma (OSCC).The following major issues should be addressed.A major postulate of this study is the specific role of RUNX2 isoform II compared to isoform I.Figure 1F shows association between patient survival and Iso II expression, but nothing is shown for Iso I, this should be added, in addition the number of patients at risk in each category should be shown.

We sincerely appreciate your valuable comments. We have added the survival curve of isoform I (exon 2.1) in the new Figure 1-figure supplement 1. In contrast to isoform II, patients with higher isoform I showed longer overall survival. The numbers of patients at risk in each category in the Figure 1F and Figure 1-figure supplement 1 are added.

The authors test Iso I and Iso II overexpression in CAL27 or SCC-9 model cell lines. In Fig. 2A in CAL27, the overexpression of Iso II is much stronger than Iso I so it seems premature to draw any conclusions. More importantly, however, no Iso l silencing is shown in either of the cell lines nor the xenografted tumours. This is absolutely essential for the authors hypothesis and should be tested using shRNA in cells and xenografted tumours.

Thank you for your valuable comments. We agree that the overexpression of isoform I is much stronger than isoform II in CAL 27 cells in Fig. 2A-B. We have done another repeat experiment which shows the similar overexpression of isoform II and I in Figure 2A-figure supplement 1. This repeat experiment also shows that overexpression of FLAG tagged isoform II significantly promoted the proliferation of OSCC cells. We tried our best to knockdown isoform I. However, the specific sequence of isoform I is 317 nt. We designed four anti-isoform I siRNAs, and unfortunately found that none of these siRNAs could knockdown isoform I efficiently. Please see following Author response image 2. Therefore, currently we cannot knockdown isoform I. However, we have tried the overexpression of isoform I. We find that isoform I overexpression inhibits the migration and invasion of cancer cells (Figure 2- figure supplement 2). In addition, we have shown that isoform II overexpression showed enhanced cell proliferation compared with isoform I overexpression in OSCC cells (Figure 2A). Therefore, we consider that isoform I is not essential for OSCC cell proliferation and tumorigenesis. Then, we mainly focus on isoform II in this study.

**Author response image 2. sa2fig2:** The knockdown efficiency of RUNX2 isoform I (anti-isoform I, si-I-1, si-I-2, si-I-3, si-I-4) in OSCC cells were analyzed by RT-PCR, 18S rRNA served as a loading control. The sequences of siRNAs are as follows: 5’ GGCCACUUCGCUAACUUGU 3’ (si-I-1), 5’ GUUCCAAAGACUCCGGCAA 3’ (si-I-2), 5’ UGGCUGUUGUGAUGCGUAU 3’ (si-I-3), and 5’ CGGCAGUCGGCCUCAUCAA 3’ (si-I-4).

A major conclusion of this study is that Iso II expression suppresses ferroptosis. To support this idea, the authors use the inhibitor Ferrostatin-1 (Fer -1). While Fer-1 typically does not lead to a 100% rescue, here the effect is only marginal and as shown in Figures 3F and G only marginally better than Z-VAD or Necrostatin 1. These data do not support the idea that the major cause of cell death is ferroptosis. Instead. Iso II silencing leads to cell death through different pathways. The authors should acknowledge this and rephrase the conclusion of the paper accordingly. Moreover, the authors consistently confound cell proliferation with cell death.

We agree that RUNX2 isoform II-knockdown could also induce apoptosis. We have revised the description in the title and abstract as follow:

Title: “RUNX2 Isoform II Protects Cancer Cells from Ferroptosis and Apoptosis by Promoting PRDX2 Expression in Oral Squamous Cell Carcinoma”.

Abstract: “In the present study, we surprisingly find that RUNX2 isoform II is a novel ferroptosis and apoptosis suppressor. RUNX2 isoform II can bind to the promoter of peroxiredoxin-2 (PRDX2), a ferroptosis inhibitor, and activate its expression. Knockdown of RUNX2 isoform II suppresses cell proliferation in vitro and tumorigenesis in vivo in oral squamous cell carcinoma (OSCC). Interestingly, homeobox A10 (HOXA10), an upstream positive regulator of RUNX2 isoform II, is required for the inhibition of ferroptosis and apoptosis through the RUNX2 isoform II/PRDX2 pathway. Consistently, RUNX2 isoform II is overexpressed in OSCC, and associated with OSCC progression and poor prognosis. Collectively, OSCC cancer cells can up-regulate RUNX2 isoform II to inhibit ferroptosis and apoptosis, and facilitate tumorigenesis through the novel HOXA10/RUNX2 isoform II/PRDX2 pathway.”.

Conclusion: “In conclusion, we identified RUNX2 isoform II as a novel ferroptosis and apoptosis inhibitor in OSCC cells by transactivating PRDX2 expression. RUNX2 isoform II plays oncogenic roles in OSCC. Moreover, we also found that HOXA10 is an upstream regulator of RUNX2 isoform II and is required for suppressing ferroptosis and apoptosis through RUNX2 isoform II and PRDX2.”.

We apologize for confusing cell proliferation with cell death. We have checked the whole manuscript and corrected the mistakes.

In Fig. 4A the authors investigate GPX1 expression, whereas GPX4 is often the key ferroptosis regulator, this has to be tested. This is important as the authors also test the effect of the GPX4 inhibitor RSL3, however, the authors do not determine IC<sub50 values of the different cell lines with or without Iso II overexpression or silencing or compared to other RSL3 sensitive or resistant cells. Without this information, no conclusions can be drawn.

We greatly appreciated the reviewer’s comments. We have performed new experiment to analyze the effect of isoform II on GPX4 expression. We find that isoform II knockdown decreases the expression of GPX4 mRNA and protein (Figure 4-figure supplement 1A and B), and conversely isoform II overexpression promotes GPX4 expression (Figure 4-figure supplement 1C and D), which is consistent with the inhibition of ferroptosis by RUNX2 isoform II. As an upstream positive regulator of RUNX2 isoform II, HOXA10 knockdown also inhibited the expression of GPX4 mRNA and protein (Figure 6-figure supplement 1A and B).

We also perform new experiment to determine IC<sub50 values of the cells with or without isoform II overexpression or silencing. We find that isoform II overexpression elevates the IC<sub50 values of RSL3 (Figure 3-figure supplement 8A), in contrast, isoform II-knockdown decreases the IC<sub50 values of RSL3 (Figure 3-figure supplement 8B).

We have added the description of these experiments in Result “RUNX2 isoform II suppresses ferroptosis”, “RUNX2 isoform II promotes the expression of PRDX2” and “HOXA10 inhibits ferroptosis through RUNX2 isoform II” as follow:

RUNX2 isoform II suppresses ferroptosis: “Isoform II overexpression could elevate the IC<sub50 values of RSL3 (Figure 3-figure supplement 8A), in contrast, isoform II-knockdown decreased the IC<sub50 values of RSL3 (Figure 3-figure supplement 8B).”.

RUNX2 isoform II promotes the expression of PRDX2: “Firstly, we found that RUNX2 isoform II-knockdown or overexpression could downregulate or upregulate the expression of GPX4 mRNA and protein, respectively (Figure 4-figure supplement 1A-D). In addition to the GPX4, we found that PRDX2 is the most significantly down-regulated gene upon isoform II-knockdown in CAL 27 (Figure 4A).”.

HOXA10 inhibits ferroptosis through RUNX2 isoform II: “In addition, HOXA10-knockdown could suppress the expression of GPX4 mRNA and protein (Figure 6-figure supplement 1A and B).”.

In summary, while the authors show that RUNX2 Iso II expression enhances cell survival, the idea that cell death is principally via ferroptosis is not fully established by the data. The authors should modify their conclusions accordingly.

We agree that RUNX2 isoform II could enhance cell survival via suppressing both ferroptosis and apoptosis. We have revised the description in the title and abstract as follow:

Abstract: “In the present study, we surprisingly find that RUNX2 isoform II is a novel ferroptosis and apoptosis suppressor. RUNX2 isoform II can bind to the promoter of peroxiredoxin-2 (PRDX2), a ferroptosis inhibitor, and activate its expression. Knockdown of RUNX2 isoform II suppresses cell proliferation in vitro and tumorigenesis in vivo in oral squamous cell carcinoma (OSCC). Interestingly, homeobox A10 (HOXA10), an upstream positive regulator of RUNX2 isoform II, is required for the inhibition of ferroptosis and apoptosis through the RUNX2 isoform II/PRDX2 pathway. Consistently, RUNX2 isoform II is overexpressed in OSCC, and associated with OSCC progression and poor prognosis. Collectively, OSCC cancer cells can up-regulate RUNX2 isoform II to inhibit ferroptosis and apoptosis, and facilitate tumorigenesis through the novel HOXA10/RUNX2 isoform II/PRDX2 pathway.”.

Conclusion: “In conclusion, we identified RUNX2 isoform II as a novel ferroptosis and apoptosis inhibitor in OSCC cells by transactivating PRDX2 expression. RUNX2 isoform II plays oncogenic roles in OSCC. Moreover, we also found that HOXA10 is an upstream regulator of RUNX2 isoform II and is required for suppressing ferroptosis and apoptosis through RUNX2 isoform II and PRDX2.”